# Fiber-based angular demultiplexer using nanoprinted periodic structures on single-mode multicore fibers

Oleh Yermakov [1,2] ✉, Matthias Zeisberger[1], Henrik Schneidewind[1], Adrian Lorenz [1], Torsten Wieduwilt[1], Anka Schwuchow[1], Mohammadhossein Khosravi[1], Tobias Tiess[3] & Markus A. Schmidt [1,4,5] ✉

Precise analysis of light beams is critical for modern applications, especially in integrated photonics, with traditional methods often struggling with efficient angular demultiplexing in compact environments. Here, we present a novel fiber-based approach that achieves angular demultiplexing through angle-sensitive coupling in nanostructure-enhanced multicore fibers. Our device uses axially symmetric nanoprinted structures to distribute the angular power spectrum of incident light over different fiber cores through higher diffraction orders. By implementing algorithmically optimized nanostructures on a seven-core single-mode fiber facet via 3D nanoprinting, we demonstrate unprecedented in-coupling efficiency over wide incident angle ranges. Our theoretical and experimental results confirm the ability of the device to function as both an angular demultiplexer and a highly efficient remote light collector. The presented approach to remotely collect and analyze light, and the combination of multicore fibers and fiber-based nanostructures, opens new possibilities for high-capacity telecommunications, environmental monitoring, bioanalytical sensing, and integrated photonic applications.

Optical fibers have become an integral part of everyday life, finding applications in areas such as optical networks, infrastructure monitoring, and life sciences. Beyond telecommunications, fibers represent an ideal platform for efficiently generating and observing light-matter interactions at remote distances. While the light transportation capabilities of fibers are exceptional, many applications require the efficient collection of light across selected angles of incidence, examples of which include endoscopy in life science[1–3] or single-photon collection in quantum technology[4,5].

Here, commercially available fibers are severely limited in their ability to collect light efficiently, especially at large angles of incidence, due to their small numerical apertures (NAs), resulting in low coupling efficiencies. For instance, in case of the incident angles larger than 20°,

commonly used all-glass single-mode step-index fibers (e.g., SMF-28[6]) show typical coupling efficiency of less than 0.0001% compared to normal incidence[7], showcasing the limitation of fibers for large-angle light collection. Note that elevating the refractive index contrast between core and cladding to increase NA includes inherent restrictions due to the material constrains. For instance, the maximum achievable NA for fused silica comercially available fiber is 0.37[8], yielding a maximum incoupling angle of 21.7°.

One promising approach to overcome this bottleneck is based on the functionalization of fiber end faces with nanostructures that influence the propagation of light through localized light-matter interaction. This approach is inline with the 'lab-on-fiber' concept[9–12] and has led to the development of complex fiber devices, including

[1]Department of Fiber Photonics, Leibniz Institute of Photonic Technology, Jena, Germany. [2]Department of Computational Physics, V. N. Karazin Kharkiv National University, Kharkiv, Ukraine. [3]Heraeus Comvance, Heraeus Quarzglas GmbH & Co. KG, Bitterfeld-Wolfen, Germany. [4]Abbe Center of Photonics and Faculty of Physics, Friedrich-Schiller-University Jena, Jena, Germany. [5]Otto Schott Institute of Material Research, Jena, Germany. ✉e-mail: oe.yermakov@gmail.com; markus-alexander.schmidt@uni-jena.de

diffractive gratings[13,14], microlenses[15,16], metasurfaces[17,18] and meta-lenses[19–21]. In addition to fabrication approaches such as modified electron beam lithography[22] or focused ion-beam milling[23], this area of research has recently been boosted by the use of 3D nanoprinting technology[16,21], which, unlike conventional wafer-based lithographic techniques, is compatible with the fiber geometry.

In the context of light coupling, nanostructures on commercial single-mode fibers (SMFs) have been extensively investigated over the last decade[11,12]. The first successful demonstrations of increasing coupling efficiency at selected incident angles were based on periodic arrays of plasmonic nanodisks[22,24], while coupling efficiencies were further boosted to percentage levels through all-dielectric axially symmetric nanostructures[25]. In addition, 3D nanoprinting technology has been used to fabricate height-adjusted non-periodic or double-periodic nanostructured ring-like gratings on SMF to facilitate coupling over large angular intervals or coupling at two angles simultaneously[7]. The approach has recently been extended to multi-mode fibers, showing improved coupling, particularly at large angles[26].

An important finding of these investigations is that purely periodic structures achieve the highest coupling efficiency, as the diffraction-based redirection of the light is based on interference. This leads to a single peak with a narrow bandwidth in the angular distribution of the coupling efficiency, which is defined by the geometric parameters of the nanostructure. Therefore, input beams outside this angular range can neither be coupled nor analyzed.

One topical area of research in Fiber Optics is multicore fibers (MCFs) that include single-mode cores, which overcome several mentioned limitations of today's SMFs. One important type of MCF is designed such that there is no crosstalk between the guided modes so that the power coupled in one core remains in that core and is not distributed across the other cores[27]. This is an excellent approach to scaling performance for different applications, examples of which include power scaling in fiber lasers[28,29], increasing data rates in telecommunications[30,31], imaging complex tissues[32,33] or realizing highly efficient sensors[34,35]. Together with the above discussion, this suggests that the combination of nanostructures and MCFs offers previously untapped advantages for light coupling into fibers.

This work presents a fiber-based concept that enables angular demultiplexing by angle-sensitive coupling in nanostructure-enhanced MCFs. By correlating specific coupling angles with the excitation of fundamental modes in different cores, the device distributes the incident angular power spectrum of an incident beam to different cores through higher diffraction orders using axially symmetric nanoprinted structures (Fig. 1). Efficient angular demultiplexers and unprecedented power collection over large incident angle intervals are demonstrated. The device is based on algorithmically optimized nanostructures using a 4-step procedure and fabricated by 3D

nanoprinting on the facets of multicore fibers with seven SMF-28-type cores. All aspects related to this concept are elucidated through detailed simulations and experimental validation.

## Results
### Multicore fiber
MCFs are a crucial component of the concept discussed in this study. They consist of seven identical cores configured in a triangular pattern, with a center-to-center spacing of $\Lambda_{MCF} = 36\,\mu m$ (Fig. 2a, b). The outer diameter of the fiber is 135 μm, thus being compatible with conventional fiber-optic circuitry. Each core, specifically engineered to mimic the modal properties of an SMF-28 fiber, operates in the single-mode regime at the operation wavelength of $\lambda_0 = 1.55\,\mu m$ (Supplementary Fig. 2). The measured mode field diameter (MFD) of the modes in the different cores is MFD $\approx 10.3\,\mu m$, leading to divergence angle of $\theta_{div} \approx 5.5°$ and numerical aperture of $NA = \sin\theta_{div} \approx 0.095$ (Supplementary Fig. 1). Note that the cores have been designed to match the corresponding values of an SMF-28 and are composed of $GeO_2$-doped fused silica, surrounded by undoped fused silica, with core diameters of approximately $d_c = 6.1\,\mu m$ (core and cladding refractive index at $\lambda_0$: $n_{core} = 1.45$ and $n_{clad} = 1.444$), see Fig. 2c, d. At the operation

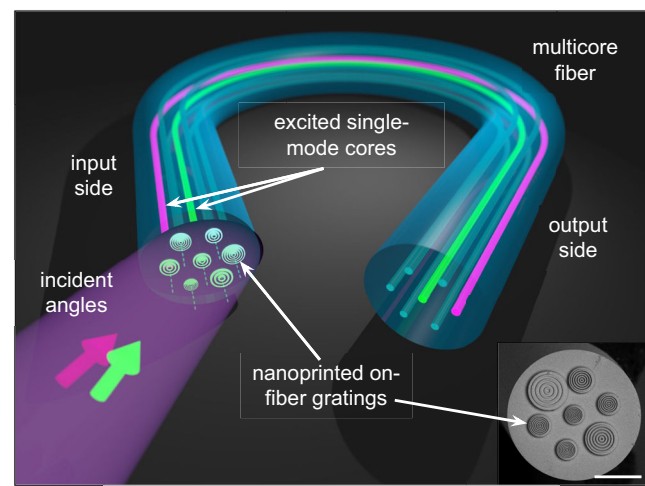

**Fig. 1 | Illustration of the concept of all-fiber integrated angular demultiplexing using nanoprinted periodic structures on single-mode multicore fibers (left: input side, right: output side).** Note that two colored arrows refer to an incident light field that includes two beams with different angles of incidence that are coupled into two selected cores. The lower right inset shows an example image of a functionalized multicore fiber that contains a nanoprinted periodic structure on each of the cores (the white scale bar corresponds to a length of 50 μm).

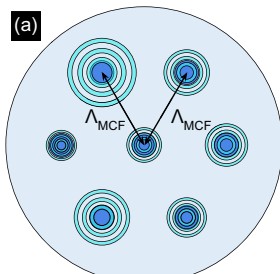 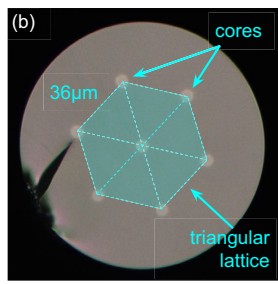 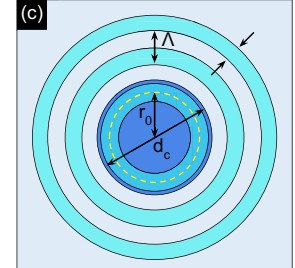 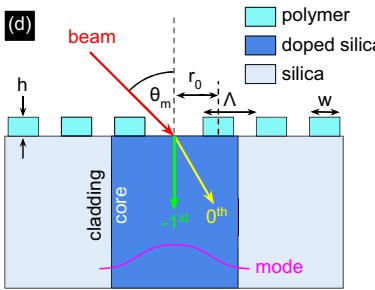

**Fig. 2 | Schematic overviews of the nanostructure-enhanced selective angle incoupling concept using single-mode MCF. a** Top view of the nanostructure enhanced MCF (dark blue: doped silica cores, light blue: silica glass, cyan: polymer). **b** Microscopic image of the surface of the bare MCF used here ($\Lambda_{MCF} = 36\,\mu m$, $d_c = 6.1\,\mu m$). The semitransparent area including the dashed lines marks the triangular arrangement of the cores inside the MCF. **c** Definition of the main

parameters of a single periodic structure (cyan: polymer, dark blue: core). The yellow dashed circle refers to the center of the first ring. **d** Schematic representation (side view) of light coupling into one core including the periodic structure (red: incident beam, green: 0th order diffracted beam, yellow: -1st order diffracted beam). The magenta line shows the intensity profile of the fundamental mode.

**Table 1 | Geometric parameters of the seven optimized designs of axially symmetric polymer structures resulting from the optimization for the different target angles ($\lambda_O = 1.55$ μm)**

| $\theta_{-1}$ | $\Lambda$ [μm] | $r_O$ [μm] | $w$ [μm] | $h$ [μm] | $N$ | $\eta_{sim}$ ($\theta_{-1}$) | $\eta_{exp}$ ($\theta_{-1}$) |
|---|---|---|---|---|---|---|---|
| 20° | 4.4 | 3 | 2.2 | 1.1 | 4 | 3.8% | 3.6% |
| 30° | 3 | 2 | 1.5 | 1.5 | 5 | 2.6% | 2% |
| 40° | 2.41 | 1.5 | 1.21 | 1.69 | 5 | 2.2% | 1.4% |
| 50° | 2.05 | 1.2 | 1.03 | 1.44 | 5 | 1.4% | 1% |
| 60° | 1.78 | 1.1 | 0.89 | 1.25 | 6 | 1% | 0.6% |
| 70° | 1.65 | 1.05 | 0.83 | 1.15 | 6 | 0.9% | 0.25% |
| 80° | 1.54 | 0.95 | 0.77 | 1.23 | 6 | 0.7% | 0.075% (0.1% at 75°) |

The two columns on the right show the values of the simulated and measured normalized coupling efficiencies.

wavelength, the modes have losses of the order of 1 dB/km (Supplementary Fig. 3). To quantify intermodal crosstalk, broadband light was injected into the center core and power transmission was measured for both the center and side cores at three bend radii, showing no crosstalk between the cores down to an extinction ratio of 25 dB (details can be found in the Supplementary Note 1, Supplementary Fig. 4). This absence of intermodal cross-coupling was additionally confirmed by coupling narrowband light (1550 nm, 10 nm bandwidth) into the center core and imaging the output with an infrared camera at different exposure times (from 1 ms to 200 ms, achieved dynamic range 46 dB), showing no evidence of intermodal cross-coupling, thus meeting the requirements of this application. Therefore, our MCFs effectively integrate the functionality of seven SMF-28 fibers into a single fiber device, resulting in a compact design.

## Main idea
The basic idea behind improving the coupling efficiency discussed here is to redirect the light by diffraction at a periodic structure on the fiber core. Previous experiments have shown that a large fraction of the electromagnetic power can be coupled into a fundamental fiber mode at large angles of incidence using the -1st diffraction order [Fig. 2d][24,25]. This requires a specific design that maximizes the in-coupling at a selected incident angle $\theta$ at the operation wavelength $\lambda_O$. The geometry followed here is a periodic array (lattice constant, i.e. pitch $\Lambda$) consisting of identical concentric rings characterized by width $w$, height $h$ and the number of rings $N$. The nanoprinted structures are being composed of a polymer with a refractive index of $n_p = 1.534$ at 1550 nm[36]. The position of the first ring, i.e. the distance from the symmetry axis to the center of the first ring, is defined as $r_0$ (Fig. 2c). This structure acts as an axially symmetric diffraction grating with a -1st diffraction order corresponding to the specific angle $\theta_m = \theta_{-1}$ [$m$: diffraction order, Fig. 2d]. Therefore, by creating gratings with different optimized coupling angles on the different cores of an MCF, the incoming angular spectrum can then be determined by analyzing the power in the different cores at the fiber output side. Note that, as shown in a previous study[25], the effect of the polarization of the incident light is practically negligible, especially within the angular ranges of the local peaks.

## Definition of fiber in-coupling efficiency
The most important benchmark parameter that determines the effectiveness of the coupling at a given angle of incidence is the coupling efficiency. This parameter describes the efficiency of converting the power of a plane wave incident on a fiber at the angle $\theta$ into one of the supported guided modes. From a practical point of perspective, the output power $P(\theta)$ is measured for different incident angles. As each structure removes a specific fraction of power from the incident

beam, all data discussed in this work is normalized to the output power of a bare fiber that contains no nanostructure at its end face at normal incidence $P_b(\theta = 0°)$, leading to:

$$\eta(\theta) = \frac{P(\theta)}{P_b(\theta = 0°)}. \qquad (1)$$

In general, $P_b(\theta = 0°)$ results in the highest possible coupling efficiency ($\eta = 1$) as (i) the direction of the incident light matches the fiber axis and (ii) any nanostructure located at the fiber tip imposes additional reflections that slightly reduces $P(\theta = 0°)$.

Another keynote value is the integrated wide-angle coupling efficiency, which is defined as follows

$$C_\eta(\Delta\theta) = \int_{\pi/2-\theta}^{\pi/2} \eta(\theta)d\theta. \qquad (2)$$

This parameter is an indicator of the accumulated total coupling efficiency over the selected angular interval $\Delta\theta = \pi/2 - \theta$ starting at oblique incidence [definition in the inset of Fig. 5d].

## Optimization strategy and structure implementation
To implement the proposed concept for distributing the angular spectrum of the incident light across the different cores of the MCF, seven different axial-symmetric nanogratings were developed using the optimization procedure described in the Methods section and Supplementary Note 2, with each of the designed gratings enabling coupling of one targeted incident angle. The angular spectra of $\eta$ for all seven nanostructure designs, both analytically and numerically optimized using the described procedure, are presented in Supplementary Note 3 (Supplementary Fig. 5). The ensemble of gratings covers an angular range from 20° < $\theta$ < 80°, with angle increments of $\Delta\theta = 10°$. The geometric data including the calculated coupling efficiency and the distribution of coupling efficiency calculated numerically are shown in Table 1 and in Fig. 3a. An exact overlap of the local maxima of the coupling efficiency with the target angles can be recognized, showcasing the relevance of the optimization procedure. Compared to reported literature[7,11,22,24], the achieved efficiencies of several percent are remarkably high and are mainly due to the discussed optimization procedure (a comparison of the various published results with those obtained in this work is presented in Supplementary Note 8). The choice of the smallest incident angle of 20° is based on the fact that at smaller angles, light is coupled into the fundamental fiber mode through the zeroth diffraction order, which leads to an increase in all curves for $\theta < 15°$. The selected angular increment of $\Delta\theta = 10°$ is sufficient to yield a negligible overlap of almost all angular response functions. Note that for $\theta > 70°$ the overlap between the curves increases due to the larger angular width of the maxima of $\eta(\theta)$, which is a consequence of the intrinsically increasing width of the first-order diffraction peaks for increasing angles of incidence (an example distribution is shown in Supplementary Note 7, Supplementary Fig. 13).

Figure 3b shows the dependence of the optimized geometry parameters (dots) on the target incidence angle $\theta_{-1}$. Clearly visible is the increase of the pitch (blue dots) towards smaller angles, which results from the intrinsic pitch behavior of a diffractive transmission grating[37] and can be approximated by the grating resonance condition used in step 1 of the optimization procedure [blue curve in Fig. 3b]. The geometry parameters $r_0$ and $w$ [green and red dots in Fig. 3b] follow this behavior and can be reproduced by slightly modified lattice resonance conditions [green and red curves in Fig. 3b]. It is important to mention that the behaviors of $r_0$ and $w$ are predetermined by their definitions in steps 2 and 3, respectively (see Methods). Note that the numerically optimized height of the grating elements oscillates around the design height parameter $h = \lambda_0/(2\Delta n)$ defined in step 3 (see

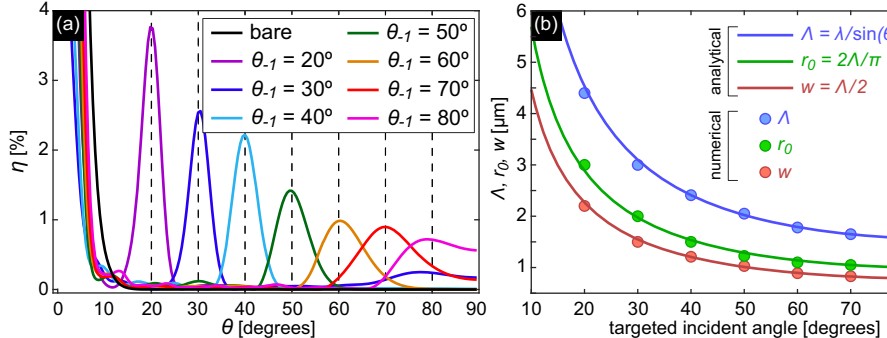

**Fig. 3 | Results of the optimization of seven axially symmetric nanogratings to excite the fundamental mode in the individual cores of the MCF at different incident angles ($\lambda_0 = 1.55\,\mu m$). a** Angular distribution of the respective coupling efficiencies. Each colored curve corresponds to an optimized angle of incidence $\theta_{-1}$, i.e. a single grating. The vertical gray dashed lines indicate the respective target optimization angle. The black curve shows the distribution in the absence of a nanostructure. **b** Dependence of the numerically optimized geometric parameters (colored dots) on targeted incident angle $\theta_{-1}$ [blue: pitch ($\Lambda$), green: distance from the center of the first ring to the origin ($r_0$), red: width of the rings ($w$)]. The solid lines were calculated using the expressions from steps 1 to 3 of the proposed optimization procedure (corresponding equations are Eqs. (3)–(5) and are given in the legend).

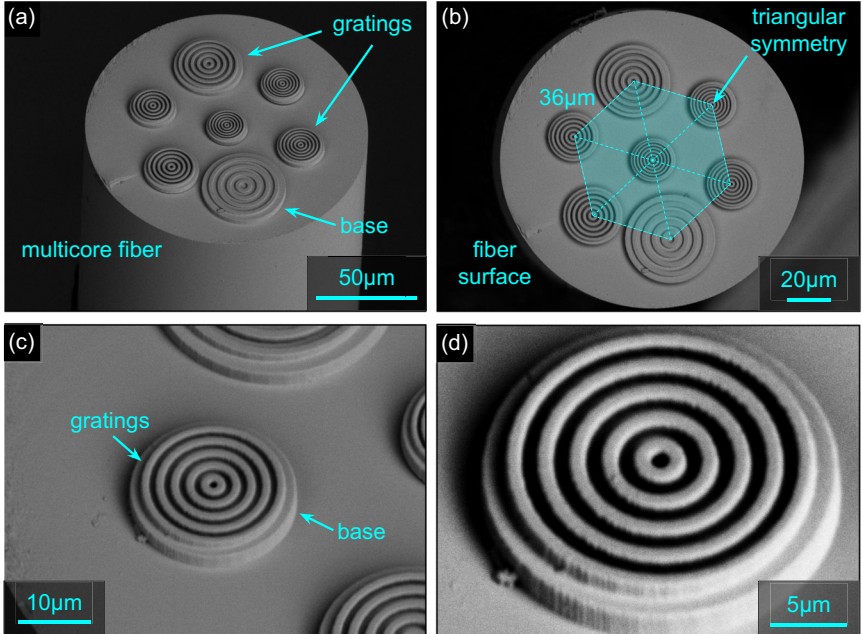

**Fig. 4 | Selected scanning electron microscope (SEM) images of the nano-printed periodic structures realized on the end face of the MCF introduced at the beginning of the work. a, b** Oblique and front views of the functionalized fiber end faces showing the printed structures at the location of the respective core. The cyan elements in (**b**) indicate the triangular arrangement of cores and gratings. **c, d** Oblique and front view of one selected nanoprinted periodic structure. Note that all structures are located on solid bases that were printed simultaneously, allowing to compensate for potential tilt of the fiber.

Supplementary Note 4). Note that, as shown by additional simulations (see Supplementary Fig. 8), variations in the height of the grating elements have minimal effect on the performance of the device, with negligible changes in the angular position and amplitude of the coupling efficiency when realistic height variations are assumed (vertical position precision 50 nm). Moreover, to demonstrate the spectral dependence of the device, the angular dependence of the coupling efficiency was simulated for the nanostructure-enhanced fiber with a design angle of $\theta_{-1} = 40°$ (grating #3 in Table 1) within the telecom C-band. The results show negligible variation in the coupling efficiency amplitude and minimal spectral shift in the maximum coupling angle of 0.02°/nm, which is practically insignificant (details can be found in the Supplementary Note 5).

The designed structures were realized on the MCFs discussed above according to the optimized parameters (Table 1) using 3D nanoprinting (SEM images in Fig. 4). The direct laser writing process used to 3D nanoprint the structures on the MCF end face is described in the Supplementary Note 6 (Supplementary Fig. 10). The 3D nanoprinting implementation procedure is described in Methods section. Note that the arrangement of the structures on the different cores was chosen to maximize the distance between the gratings, so that the largest structures are outside the central region of the MCF. Visual inspection of the images obtained shows that the nanoprinted structures reproduce their simulated counterparts to a very high degree.

## All-fiber integrated light collector

The experimental characterization of the coupling efficiency for each core of the nanostructure-enhanced MCF (Fig. 5a, corresponding peak values in Table 1) shows distinct peaks in the angular distributions that match the target angles $\theta_m$. For $\theta \leq 50°$, the values of the in-coupling efficiency $\eta$ reach percentage level, while for $50° \leq \theta \leq 80°$ the coupling

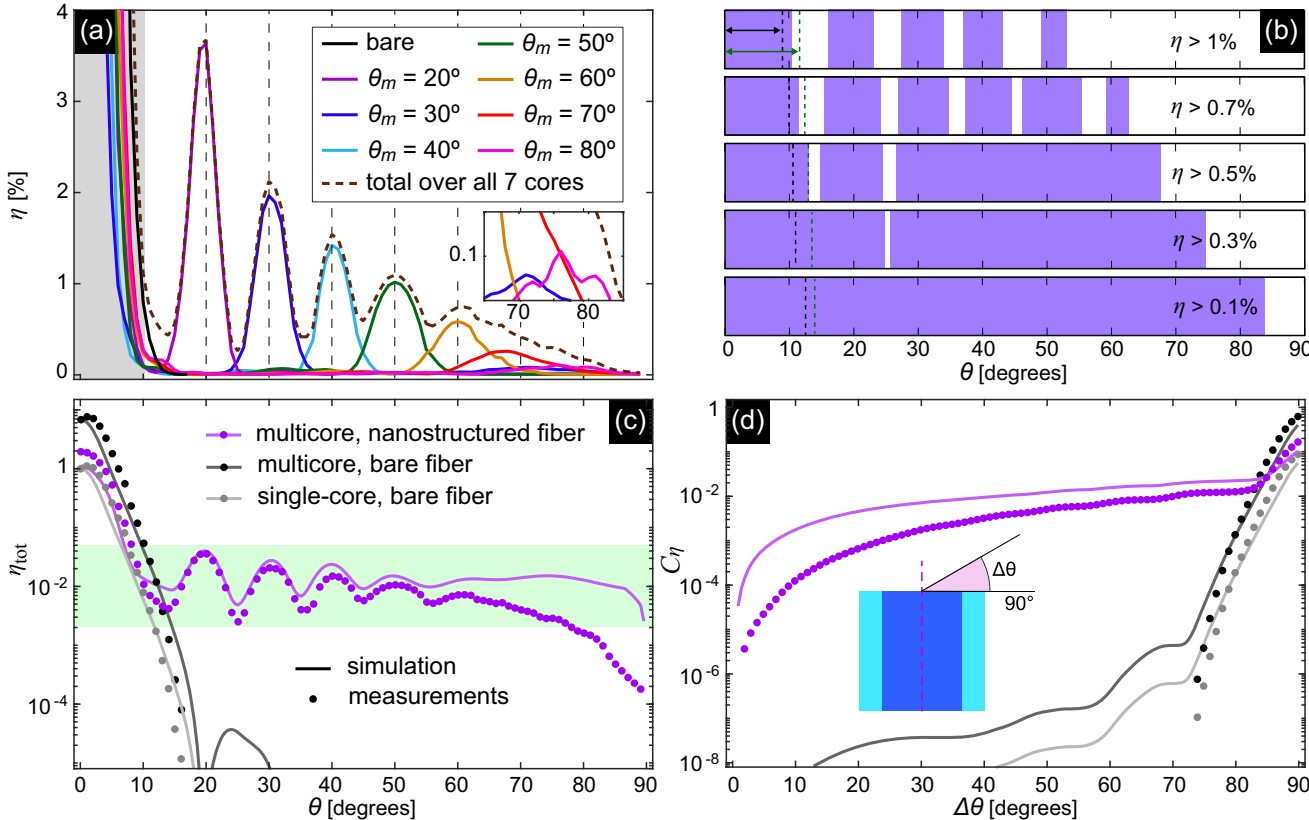

**Fig. 5 | Results of the optical characterization. a** Measured coupling efficiencies of the individual cores containing nanostructures (colored curves) compared to a blank core (black line). All curves are normalized to the coupling efficiency of a single bare core at normal incidence (Eq. (1), $\theta = 0°$, $\lambda_0 = 1.55\,\mu m$). The corresponding total coupling efficiency over all seven structured cores $\eta_{tot}$ is shown by the brown dashed line. The vertical gray dashed lines indicate the targeted optimization angles $\theta_{-1}$ overlapping with local maxima of $\eta$ for each nanostructured core. The angular domain $\theta < 10°$ in which the 0th-diffraction is dominantly coupled into the cores is highlighted by the light gray background. The inset shows the angular interval from 65° to 85°, highlighting the coupling at almost grazing incidence. **b** Angular intervals (purple areas) in which the total coupling efficiency

exceeds a predefined value [0.1% (bottom) to 1% (top)]. The black and green vertical dashed lines indicate the corresponding limits for bare single and multicore fibers. **c** Angular distribution of the measured total coupling efficiency (dots: experimental measurements, lines: numerical calculations) of nanostructure-enhanced MCFs (purple) in comparison to an MCF without nanostructures (dark gray). The light gray curve shows the corresponding distribution for a single blank core. **d** Integrated wide-angle total coupling efficiency defined in Eq. (2) of nanostructured-enhanced (purple) and bare MCF (dark gray) in comparison to a single core (light gray). The inset illustrates the definition of the large-angle interval (dots: experiments, lines: simulations).

efficiencies are about 0.1–1%. It is noteworthy that the coupling efficiency at almost grazing incidence ($\theta \approx 75°$) is 0.1%, which is an order of magnitude higher than that of a blank fiber at a significantly smaller angle (e.g., $\theta = 15°$).

As each of the nanostructure-enhanced cores is able to collect light outside its peak region, a total coupling efficiency of the entire device can be defined. This parameter is given by the sum of the measured coupling efficiencies of all cores at each possible angle ($\eta_{tot}(\theta) = \sum \eta_j(\theta)$; $j = 1. . . 7$) and reflects the integrated coupling efficiency at the output of the MCFs [brown dashed line in Fig. 5a]. The improved coupling is clearly recognizable at all incident angles, even between two peaks and especially at large angles ($\theta \geq 60°$).

Figure 5b shows the angular intervals in which the total coupling efficiency exceeds certain predefined values. For example, $\eta_{tot}$ is greater than 0.1% and 0.3% in the whole angular range up to maximum angles of 83° and 74°, respectively. At the same time, up to 68° and 63° $\eta_{tot}$ is almost always > 0.5% and > 0.7%, respectively. Percentage levels of coupling efficiency are reached in angular intervals up to 53°, showing an oscillatory modulation of $\eta_{tot}$ [Fig. 5a, c]. Considering an MCF with a bare interface, the intervals of efficient in-coupling are significantly limited [vertical green dashed lines in Fig. 5b]. Here, total in-coupling efficiencies greater than 1%, 0.5% and 0.1% are only achieved up to angles of about 12°, 13° and 14°, respectively. A similar

behavior at even smaller angles is found when considering the power from one single core only [vertical black dashed lines in Fig. 5b].

A more detailed comparison of the angular distribution of the total coupling efficiency between nanostructure-enhanced and bare MCFs is shown in Fig. 5c. It is evident that the nanoprinted structure significantly improves the total coupling efficiency for $\theta > 13°$ and especially at larger angles. Specifically, $\eta_{tot}$ for a bare MCF falls below $10^{-6}$ when $\theta > 17°$ while for the nanostructure-functionalized MCF, $\eta_{tot} > 10^{-4}$ over the entire angular range. Note that the light collection efficiency of the functionalized MCF exceeds that of its unstructured counterpart by 2 ($\theta > 16°$) to 4 ($\theta > 30°$) orders of magnitude. Both experimental and numerical results are in good agreement, especially for $\theta < 60°$. The nanoprinted polymer structures reduce the coupling efficiency at normal incidence, indicating that the device is designed to operate predominantly in the non-normal incidence range (Supplementary Table 3 shows the coupling efficiency of all configurations).

The improvement in light collection, especially at large angles, becomes even more evident when the integrated wide-angle coupling efficiency $C_\eta$ is considered [see its definition in Eq. (2) and the inset of Fig. 5d]. Here, the functionalized MCF shows a significantly better performance by orders of magnitude than the bare fiber over a very wide angular range [$\Delta\theta > 70°$, Fig. 5d].

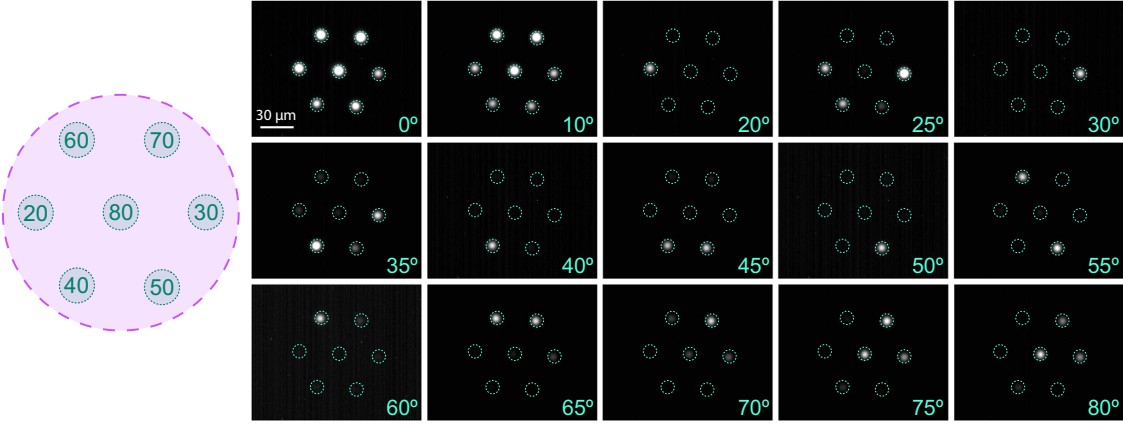

**Fig. 6 | Selected measured images of the output intensity of the nanostructured-enhanced multicore fiber (Supplementary Movie 1), showing the power distribution across the different cores at different angles of incidence (given at the bottom of each image, $\lambda_0 = 1.55\,\mu m$).** In each image, the domains of the fiber cores are indicated by dashed cyan circles. The optimized angles ($\theta_{-1}$, in unit of degree) together with the associated core are indicated in the schematic on the left. Note that each image was captured with a different gain factor to compensate for the limited dynamic range of the camera used (a table of the corresponding gain factors is provided in Supplementary Note 6).

## All-fiber integrated angular demultiplexer

The main feature of the MCF approach over a single-core fiber is the ability to distribute a power spectrum of incident angles across different cores, resulting in a fiber-based angular demultiplexer. To demonstrate and reveal the limitation of the concept, the angle of incidence was continuously varied from 0° to 90° and for each angle an image of the spatial power distribution at the output of the nanostructure-enhanced MCF was taken (a selection of 15 measured images is shown in Fig. 6).

For small incident angles up to $\theta < 10°$, the power in all seven cores appears to be identical, which is associated with the coupling of the 0th diffraction order (gray area in Fig. 5a). Above this angle, the incident angular power spectrum is spatially separated according to the assignment of the cores to the corresponding optimized gratings. In detail, if the angle of incidence corresponds to one of the targeted optimized angles ($20° < \theta_{-1} < 80°$ in steps of $\Delta\theta = 10°$), one of the cores appears dominant in the related image. This is the result of the seven well-separated maxima of the coupling efficiency [colored curves in Fig. 5a], each of which imposes coupling in a selected core using the optimized nanostructures (c.f. Table 1). Even more interesting is the case of an incident angle located between two adjacent target angles $\theta_{-1}$, resulting in 2 or 3 cores emerging in the images at an overall lower level of output power (Fig. 6 shows 13 cases between 20° and 80°). This effect can be clearly seen in Supplementary Movie 1, which shows the dynamic change of the power distribution across the different cores in case the angle of incident is scanned. A potential application scenario that often arises in the context of near-infrared light sources is the identification of an unknown direction of a light source. It is important to note that such a determination using the fiber-based device discussed requires a comparison of the power levels in different cores, as a single-core measurement does not provide sufficient information. Specifically, the procedure involves (i) identifying the core with the highest output power, which corresponds to one of the colored curves in Fig. 3a, and (ii) comparing the power levels of modes from several related cores, which allows the angle of incidence to be uniquely determined. An example of this procedure is shown in Supplementary Note 5.

## Discussion

In conclusion, our study presents a novel fiber-based concept that demonstrates angle demultiplexing via angle-sensitive coupling in nanostructure-enhanced multicore fibers. By correlating specific coupling angles with the excitation of fundamental modes in different cores, the power spectrum of an angular distribution is distributed across different cores. The device can either act as (i) an efficient angular demultiplexer or (ii) allows for a record high collection efficiency over unprecedented large incident angle intervals. The key feature of the device is the integration of two essential components: (i) Algorithmically optimized, axially symmetric polymeric periodic structures fabricated by advanced 3D nanoprinting, representing the first time 3D nanoprinting has been applied to MCFs, effectively utilizing higher diffraction orders to direct light into specific cores. This study also addressed the challenge of precisely positioning multiple nanostructures at specific locations on the fiber endface. (ii) Multicore fibers with seven single-mode cores that exhibit no modal crosstalk and a very high degree of integration via integrating the functionality of seven SMF-28 fibers into a single fiber. The study includes detailed theoretical analysis, introduces a novel computational optimization approach that is based on a four-step procedure, experimental implementation via fiber drawing and 3D nanoprinting, and detailed optical characterization, which together highlight the advantages and limitations of the concept.

The presented approach to collect and analyze light as well as the combination of multicore fibers and fiber-based nanostructures as a whole are promising for a wide range of applications including wavelength-division-multiplexing in high-capacity telecommunication networks, trace gas detection in environmental sciences, molecular sensing in bioanalytics, and efficient light coupling within integrated photonics and nano-optics. The concept is particularly well suited for fiber-based spatial multiplexing systems, where the angular power spectrum can be distributed over multiple spatial channels for efficient light collection and routing, and can particularly benefit from highly dispersive nanostructures such as metasurfaces[38] or photonic crystal-based superprims[39]. It is important to highlight that the fabrication process employed generally allows for the creation of structures with significantly greater complexity, thereby substantially broadening the spectrum of potential applications. Here, advances in fiber end face functionalization (e.g., laser structuring of graphene layers[40], focused ion-beam milling[23], modified electron beam lithography[22]) or conceptually innovative approaches (e.g. focussing via higher-diffraction orders[41]) may open new opportunities to create complex on-fiber devices with unique properties, in particular for efficient light coupling.

Overall, the technological advances presented in this study include the integration of cutting-edge technologies that combine innovative fiber drawing techniques with high-precision 3D nano-printing to fabricate a monolithic fiber-based device that includes a novel functionality. A conceptually novel approach to measure the

angle distributions of incident beams by correlating specific coupling angles with the mode of a specific core is introduced, along with a computational design optimization procedure. Note that the device achieves record-high light collection efficiencies over previously unattainable wide incident angle intervals.

The concept presented is based on axially symmetric diffraction gratings designed to couple light into a specific core within a defined angular interval. One direction for future optimization is to explore alternative grating types. For example, Dammann gratings modified by algorithmic optimization have been shown to effectively suppress the zeroth order of diffraction while selectively directing light output to higher diffraction orders[42]. In addition, complex three-dimensional structures such as photonic crystal-based superprisms[39] could potentially be applied directly to fibers using 3D nanoprinting for redirecting the light. Another topical approach is the use of metasurfaces or phase holograms, which allow precise phase manipulation to control light[20,21,43]. Artificial intelligence is also expected to play a role in refining these structures to improve their performance[44,45]. In addition to the aforementioned benchmarks–maximizing and suppressing the 0th order diffraction–other goals include equalizing coupling efficiency across all angles and narrowing the bandwidth of individual peaks to simplify data analysis and to increase channel density.

A key aspect of the approach discussed is the use of single-mode MCF, which shows no intermodal crosstalk for the fiber lengths of interest here (<10 m). One approach to improve performance is to increase the number of cores in order to reduce the angular increment between the target angles $\Delta\theta$. Recent advances in fiber fabrication techniques show fibers with significantly reduced modal crosstalk at higher core densities, for example by embedding individual cores in a material with suppressed refractive index[46]. Another strategy is to use shorter operation wavelengths, which intrinsically lead to reduced mode coupling and thus a higher core density. In addition, the use of fibers with thermally expanded cores is a viable method of increasing the incoupling efficiency through larger structures. In this technique, a heat source is used to induce local diffusion of $GeO_2$ molecules from the core into the surrounding cladding, thereby spatially broadening the mode and providing a larger interaction area between incident light and the nanostructure. This approach has been shown to be highly effective for single-mode fibers[47,48] also in combination with nanoprinted metalenses[49] and recent developments indicate its applicability to multicore fibers[50].

## Methods

### Optimization procedure for nanostructure design
A critical requirement for the fiber device described here is that each of the seven cores must be assigned a specific coupling angle with the highest possible coupling efficiency. To achieve this, an optimization procedure has been developed, which allows maximizing the excitation of a mode in a specific core for a given coupling angle. This procedure allows finding optimal grating parameters and consists of the following four main steps. Here, steps 1 to 3 lead to a purely analytic-based design, while step 4 conducts a final numerical optimization. The detailed derivation and explanation of each step is presented in Supplementary Note 2.

Step 1: Calculation of the lattice constant (i.e. pitch) via the lattice equation[22]:

$$\Lambda = \lambda_0 / \sin \theta_m. \tag{3}$$

Step 2: Determination of the position of the center of the first ring via:

$$r_0 = 2\Lambda/\pi. \tag{4}$$

Step 3: Determination of width and height of the rings, being identical for all rings considered:

$$
\begin{aligned}
w &= \Lambda/2, \\
h &= \lambda_0/(2\Delta n),
\end{aligned}
\tag{5}
$$

where $\Delta n = n_p - n_{air}$ is the difference between the refractive indices of polymer and air.

Note that the analytically determined heights result from the condition that under normal incidence, the phase difference between the wave passing through the polymer and the air domains is equal to $\pi$, which implies that all elements have the same height in this step.

Step 4: Numerical optimization of the coupling efficiency via Finite-Element modeling, considering the previously determined parameters as starting values.

To showcase the capabilities of the optimization approach, a polymer periodic structure has been designed for improving the coupling into one core of the MCF, targeting a coupling angle of $\theta_{-1} = 40°$ (the full results are shown in Supplementary Note 3).

### 3D nanoprinting
The structures were fabricated using a commercially available nano-printer (Photonic Professional GT2, Nanoscribe GmbH, Germany) employing two-photon polymerization with a resolution of 300 nm × 300 nm × 1000 nm (further details can be found in ref. 7). Optimized printing parameters allowed for the creation of seven individual annular gratings on the fiber's end face, each aligned with a selected core. During development, the unpolymerized resin was removed by first immersing the fiber end in propylene glycol monomethyl ether acetate (PGMEA) for 20 min, followed by a 3-min rinse in nonafluoro-(trifluoromethyl)-pentanone (Novec).

### Optical measurement procedure
Optical measurements were performed by illuminating the nanostructured end of the sample with collimated light ($\lambda_0 = 1550$ nm) and capturing the output pattern with a camera (c.f., Supplementary Note 6 and Supplementary Fig. 11). Note that the entire fiber end face is illuminated. To compensate for the limited dynamic range of the camera, images were taken with several exposure times at the same angle if necessary (the related measurement procedure is shown in Supplementary Fig. 12). The angular distribution of the coupling efficiency was determined by rotating the MCFs, mounted on a high-precision rotary stage (accuracy is 0.5°), relative to the input beam.

### Data analysis
The evaluation of the recorded data included image processing using ImageJ software, which includes integration of gray values over a predefined area for each core. To improve signal quality, particularly at low intensities, the noise contribution in a similar area adjacent to the fiber cores was determined and subtracted from the signal. For comparison, a bare MCF characterized in the same setup yields 37.4 µW of total output power from all cores combined under normal incidence.

## Data availability
All data that support the findings of this study are present in the paper and the Supplementary Information. All raw data generated during the current study are available from the corresponding authors upon request.

## Code availability
We employed the POV-Ray (Persistence of Vision Raytracer, version 3.7) framework[51] to produce Fig. 1. The codes used in the current study are available from the corresponding authors upon request.

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

## Acknowledgements

The multicore fiber was created in the BMBF project SAMOA-Net, funding code 16KIS1424. The authors acknowledge the support from the German Research Foundation (DFG) via the grants SCHM2655/21-1, SCHM2655/23-1. O.Y. acknowledges the support of the Alexander von Humboldt Foundation within the framework of the Humboldt Research Fellowship for postdoctoral researchers and the American Physical Society's International Research Travel Award Program-Ukraine (IRTAP-Ukraine) with funding from the Alfred P. Sloan Foundation.

## Author contributions

A.L., T.W., A.S., M.-H.K. and T.T. manufactured and experimentally characterized the multicore fiber. M.Z. fabricated the nanostructures on multicore fiber tip using 3D nanoprinting. H.S. carried out the measurements of the nanostructured multicore fiber. O.Y. and M.S developed the analytical model and optimization strategy of the nanostructures on fiber tip. O.Y. carried out the numerical calculations. O.Y. and M.S. wrote a first version of the manuscript that was edited by the rest of the authors. M.S. supervised the work and conceived the original idea for this research.

## Funding

## Competing interests

The authors declare no competing interests.
