## [Transparent Peer Review file · Nature Communications]

Fiber-based angular demultiplexer using nanoprinted periodic structures on single-mode multicore fibers

Corresponding Author: Professor Markus Schmidt

Version 0:

Reviewer comments:

Reviewer #1

(Remarks to the Author)

The authors have demonstrated an all-fiber integrated approach for demultiplexing light at different incidence angles using nanoprinted structures on the end face of a multicore and single-mode fiber. A large incident angle and angle-sensitive coupling can be achieved by optimizing the nanostructures on each fiber core. This study includes both simulation and experimental data to support their proposed technique. Although the presented nanoprinting technique on the MCF tip looks interesting, below are my detailed concerns and comments.

1. Their previous studies have already shown the enhancement of the fiber numerical aperture through nanostructures, e. g. Plidschun, et al., *Light Sci. Appl.* 10(1), 1–11 (2021); Zeisberger, M., et al. *Opt. Lett.* 49(8), 1872–1875 (2024). When this is combined with an increased number of fiber cores each having unique nanostructures, the improvement in incident angle appears to be achievable without significant technological advancements.
2. In the proposed method utilizing multiple fiber cores with nanostructures, the collimated beam size should be sufficiently large to "illuminate the entire fiber end face" and cover 7 fiber cores. In comparison to previous methods that utilize single-mode fibers, is there a risk of a significant decrease in efficiencies with the proposed method? It appears from Figure 1 that adjustments may also be required on the input side to position the beam accurately on each fiber core, rather than relying on a large beam that can cover the entire fiber end face.
3. The authors mentioned that "To eliminate the power dependence on the intensity of the incident light, all data discussed in this work is normalized to the output power of a bare fiber that contains no nanostructure at its end face at normal incidence". Why does the incident light in the experiment exhibit varying intensities? It would be beneficial to provide the normal incidence efficiency $P_b(\theta=0)$ for readers to understand the potential efficiency achievable with multicore fibers.
4. "Compared to reported literature [7, 11, 22, 23], the achieved efficiencies of several percent are remarkably high and are mainly due to the discussed optimization procedure." It is recommended to include a comparison table to provide a comprehensive and fair assessment of the enhancements facilitated by their method.
5. There is a figure reference error with corrupted characters in Section 4.1.

Reviewer #2

(Remarks to the Author)

The present manuscript presents an all-fiber integrated angular demultiplexer by printing periodic structures on a multicore fiber tailored to couple light at specific angles. The present work shows an alternative of a microstructure on the tip of an optical fiber that allows to selectively couple light with different angle of incidence into different fiber cores. The work presents an integrated solution to capture or emit light from angles where usually conventional optical fibers are limited. The impact of this work seems mostly for free-space applications such as communications, spectroscopy, or integrated photonics, for just mentioning the most relevant ones. Although structures are simple and design is not very complicated, it does require a very specialized lithography system; nonetheless, the progress in this area should simplify the fabrication of such structures. The manuscript is well written and the structure of the article is appropriate. I recommend the article for publishing but after some aspects are addressed.

A major aspect missing is how the structure works for a light source with a light source of broadband spectrum. The only example of this can be found in Fig. 3b, but no further comments or analysis are provided. For instance, calculations of how much the coupling efficiency changes for the structure within a short wavelength range, such as 5 or 10 nm wavelength span. Another example would be to extend the analysis of coupling efficiency for the C-band, since in the application it is mentioned that it can potentially be used for optical communications.

In Fig. 6 the distribution of the periodic structures for the different angles is presented. How were the different structures corresponding for different incident angles assigned? Is there an optimal distribution or some guidelines for this? This is a major aspect since the output patterns of the MCF for light launched from different angles does couple to different cores, especially at small angles. How can this be minimized and please provide some numbers of "crosstalk" for the proposed device. For example, for an unknown direction light source (1550 nm), what is the relationship for the intensity output at different cores and how the incidence angle can be determined? Additional discussion on the selectivity of the device must be included.

Since the structures are symmetrical, polarization dependence is not expected. However, this is not mentioned. How does the polarization of light interacting with the structure play a role?

In Table 1 of the supplementary material it can be observed that the error in the heights of the periodic structures is big while the designed and optimized structure are compared. Why is this? In addition, the heights of these structures need fabrication resolutions of 10 nm. What is the resolution of the fabrication process in height and what are the penalties in the response of the structures if a resolution of 0.1 μm is only achievable?

In the supplementary material, section S1, the characterization of the MCF is explained. The cores of the MCF were designed to not present coupling among the cores. Although bending losses are nicely characterized, how the bending affects the mode coupling among cores is not shown in this study. This is important because the fiber design is part of this manuscript. An explanation of this aspect and some calculations should be included.

It contains detailed information regarding the design of the periodic structures. More detailed information is provided in the supplementary material. However, during the reading of the main article sometimes it is not clear that all the design and characterization is performed at a wavelength of 1550 nm. This must be mentioned in all the plots and tables.

As the periodic structures are fabricated on top of each SM core and are designed to capture light at different incidence angle, I missed how the losses are due to the periodic structures while the incidence is normal. Please clarify this.

The structures are demonstrated on a MCF, but this type of application should work as well on all-fiber spatial multiplexers. Please include this in the discussion of the manuscript.

References cited in the manuscript are adequate but some other works with similar structures should be included and commented:

- <https://pubs.acs.org/doi/10.1021/acs.nanolett.0c04463>
- <https://doi.org/10.1038/s41467-024-52256-y>
- <https://doi.org/10.1038/s41598-017-04490-2>
- <https://doi.org/10.1364/PRJ.521005>
- <https://doi.org/10.1038/s41467-022-31902-3>
- <https://doi.org/10.1038/s41377-023-01222-2>

Some minor aspects are:

- In Fig.5, a) is not defined in the description of the figure.
- In the paragraph below Table 2, some Figures are mentioned but it seems that the PDF file was not correctly compiled. This
- In the first paragraph of section 4.2, the reference cited is not correctly compiled. This happens as well in section S3.

Reviewer #3

(Remarks to the Author)

This study presents an angular demultiplexer achieved by applying nanoprinted periodic structures onto the facet of single-mode multicore fibers. Light with a large incident angle, significantly exceeding the acceptance angle of the single-mode fiber (SMF), can be redirected through diffraction at the periodic structure on the fiber's surface, enabling the light coupled into the core of single-mode multicore fibers. The author demonstrated the in-coupling efficiency exceeds 1% for incident angles between 20 to 50 degrees and 0.1% to 1% for angles between 50 and 80 degrees. This is an interesting result that deserves to be published. I have however several critical points that must be revised prior to publication.

1. The paper claims this device is "all-fiber integrated", which usually means the device splice robustly with other fiber components. i. e. Sohanpal, Ronit, et al. "All-fibre heterogeneously-integrated frequency comb generation using silicon core fibre." (2021). The device used in this work was fabricated by printing nano periodic structures on the facet of multicore fibers and has not been spliced with other fiber component. It is not sensible to use "all-fiber integrated".
2. It would be better to have schematic figures for the 3D nanoprinting and optical measurement procedure, which would produce more details and could be helpful for other labs to follow and reproduce this work after being published.
3. In Figure 6: why in the beam in the fig of 25 degree is brighter than 30 degree and the fig of 35 degree is brighter than 40 degree. Is this caused by the normalization to a maximum signal-to-noise ratio at the specific angle of incidence? It would be

better to use the same normalization level, and consistent with the video this paper produced.

4. Section 4.1 “the green curves in Figs. ??g-??h ” and “blue curves in Figs. ??g-??h ” please fix the editing errors.

Version 1:

Reviewer comments:

Reviewer #1

(Remarks to the Author)

The authors have addressed all of my concerns. I have no further questions.

Reviewer #2

(Remarks to the Author)

The Authors have successfully addressed thoroughly and meticulously all the points after the first revision. The additional material and results included in the new manuscript resulted in a nice work that deserves to be published. I thank the Authors for extending the experimental work and simulations to complement the submitted manuscript. As a result, the new version of the manuscript reads very well and explains in detail the work. Also, as previously mentioned, the quality and importance of the work is very high. Therefore, I fully recommend that the article be published. I only have a few comments.

The addition of Supplementary Figure 4 is a nice addition to the content of the manuscript. A suggestion that will improve the Figure is to also illustrate with dashed-line circles where the other cores should be; this will also help to avoid misunderstandings due to the artifact visible in the images.

In the revised manuscript, I observed on page 14 that the Authors mention Figs. S4g-S4h which I believe are incorrect. After the new additions to the manuscript it is now S5g-S5h. As new material (tables and figures) were added, please check the numbering and mentions of those.

The first part of the question previously done (R2.2) the intention was more to know if the distribution of the different structures had a specific order. Regarding the design of the structures, the original version of the manuscript was already well described. Nonetheless, with the information added regarding the questions from reviewer 1, it is clear that the light is intended to impinge on the whole fiber facet of the fiber. Thus, this is clarified.

Reviewer #3

(Remarks to the Author)

The authors have addressed and responded to all of my comments in high quality. I have no further comments on the revised manuscript and recommend this work be published in Nature Communications.

Prof. Dr. Markus A. Schmidt

Leibniz Institute for Photonic Technologies (IPHT),
Albert-Einstein-Straße 9, 07745 Jena, Germany

Otto Schott Institut of Material Research (OSIM)
Friedrich Schiller University Jena
Fraunhoferstr. 6, 07743 Jena, Germany

T: +49 (0) 3641 206 140
markus.schmidt@ipht-jena.de

13 December 2024

Response letter for our manuscript "*Fiber-based angular demultiplexer using nanoprinted periodic structures on single-mode multicore fibers*" (ID: NCOMMS-24-45639)

Dear Reviewers,

Please find attached the revised version of our manuscript entitled "*Fiber-based angular demultiplexer using nanoprinted periodic structures on single-mode multicore fibers*" (old title: "*All-fiber integrated angular demultiplexer using nanoprinted periodic structures on single-mode multicore fibers*") by Oleh Yermakov, Matthias Zeisberger, Henrik Schneidewind, Adrian Lorenz, Torsten Wieduwilt, Anka Schwuchow, Mohammad-Hossein Khosravi, Tobias Tiess, and Markus A. Schmidt. We are grateful to the Reviewers for the feedback and for raising important points, which have helped us to improve the manuscript.

Please note that we have added Mohammad-Hossein Khosravi to the list of authors as he performed additional experiments (i.e., measurements of intermodal coupling) necessary to address one of the reviewer's comments, which are included in the Supplementary Information.

In the following, the Reviewer comments are given in black. Please find our responses in blue and the revised text in green (changes are highlighted in green bold). Please note that all changes (except minor spelling corrections) are detailed in this response letter, including the locations where they appear in the manuscript.

We take the opportunity to thank the Reviewers once again for their constructive comments and suggestions, which helped us to substantially improve the quality of our manuscript. Having responded to their queries in detail, we look forward to your response.

Sincerely,

Markus A. Schmidt

Response to comments of Reviewer #1

The authors have demonstrated an all-fiber integrated approach for demultiplexing light at different incidence angles using nanoprinted structures on the end face of a multicore and single-mode fiber. A large incident angle and angle-sensitive coupling can be achieved by optimizing the nanostructures on each fiber core. This study includes both simulation and experimental data to support their proposed technique. Although the presented nanoprinting technique on the MCF tip looks interesting, below are my detailed concerns and comments.

We sincerely thank the Reviewer for the thorough review of our manuscript and the detailed comments. Below we have provided detailed responses to each of the Reviewer's comments. We hope that our revisions have addressed all concerns and contributed to a stronger and higher quality version of the manuscript.

R1.1: Their previous studies have already shown the enhancement of the fiber numerical aperture through nanostructures, e. g. Plidschun, et al., Light Sci. Appl.10(1), 1–11 (2021); Zeisberger, M., et al. Opt. Lett. 49(8), 1872–1875 (2024). When this is combined with an increased number of fiber cores each having unique nanostructures, the improvement in incident angle appears to be achievable without significant technological advancements.

We thank the Reviewer for highlighting the important point of technological advancement. While we acknowledge that the fabrication of dielectric nanostructures on fiber end faces using 3D nanoprinting has been reported in literature and demonstrated by our group, this work represents the first demonstration of the combination of a multicore fiber with a nanoprinted structure for an application that is highly relevant in the context of Fiber Optics, thus including both a substantial technological and a conceptual advancement. Below we present these aspects in detail, divided into technical and conceptual novelty.

Technological novelty

- **Implementation of novel device functionality:** First time demonstration of a novel fiber-integrated device for angular demultiplexing by angle-sensitive coupling into the different cores of nanostructure-enhanced multicore fibers (MCFs), effectively exploiting tailored higher diffraction orders for targeted light coupling into specific fiber cores.
- **First time 3D nanoprinting on multicore fiber:** Realization of algorithmically optimized periodic structures on MCFs using advanced 3D nanoprinting. Beyond printing, this study also addressed the technological challenge of precisely positioning multiple nanostructures at specific locations on a fiber surface.
- **Realization of cross-talk free SMF-28-like multi-core fiber:** Design and implementation of a MCF with seven single-mode cores without modal crosstalk, combining the functionality of seven standard telecom fibers into one MCF.
- **Efficient collection over large angular intervals:** Record-high light collection efficiencies over previously unattainable wide incident angle intervals.
- **Integration of cutting-edge technologies:** Combination of innovative fiber drawing techniques with high-precision 3D nanoprinting to create a fully integrated, monolithic fiber-based device.

Conceptual Novelties

- **New concept for measuring angle distributions of incident beams via fully fiber-integrated device:** Development of a novel concept to measure the angular distribution of an incident beam by correlating specific coupling angles with the excitation of fundamental modes in selected cores, enabling a spatially resolved representation of angular distribution information.
- **Introduction of a fully integrated monolithic device:** Introduction of an unexplored fully integrated monolithic fiber device capable of unprecedented angular demultiplexing and record-breaking light collection efficiency over large angular intervals.
- **Novel design optimization approach combined with extensive theoretical analysis:** Introduction of a novel computational approach to nanostructure design optimization, integrated with detailed theoretical analysis and optical characterization, laying the groundwork for future design improvements.

Based on these aspects, we believe that the work, and in particular the concept of angular demultiplexing through angle-sensitive coupling in nanostructure-enhanced MCFs, contains a substantial amount of technological and concept novelty and advancement.

To address the Reviewer's comment, the text of the Discussion and Conclusions section of the manuscript has been changed as follows:

... The key feature of the device is the integration of two essential components: (i) Algorithmically optimized, axially symmetric polymeric periodic structures **fabricated by advanced 3D nanoprinting, representing the first time 3D nanoprinting has been applied to MCFs, effectively utilizing higher diffraction orders to direct light into specific cores. This study also addressed the challenge of precisely positioning multiple nanostructures at specific locations on the fiber endface.** (ii) Multicore fibers with seven single-mode cores ...

... It is important to highlight that the fabrication process employed generally allows for the creation of structures with significantly greater complexity, thereby substantially broadening the spectrum of potential applications.

Overall, the technological advances presented in this study include the integration of cutting-edge technologies that combine innovative fiber drawing techniques with high-precision 3D nanoprinting to fabricate a monolithic fiber-based device that includes a novel functionality. A conceptually novel approach to measure the angle distributions of incident beams by correlating specific coupling angles with the mode of a specific core is introduced, along with a new computational design optimization procedure. Note that the device achieves record-high light collection efficiencies over previously unattainable wide incident angle intervals. ...

R1.2: In the proposed method utilizing multiple fiber cores with nanostructures, the collimated beam size should be sufficiently large to "illuminate the entire fiber end face" and cover 7 fiber cores. In comparison to previous methods that utilize single-mode fibers, is

there a risk of a significant decrease in efficiencies with the proposed method? It appears from Figure 1 that adjustments may also be required on the input side to position the beam accurately on each fiber core, rather than relying on a large beam that can cover the entire fiber end face.

We thank the Reviewer for raising this important point. First, we would like to mention that, as the Reviewer pointed out, the entire cross-section of the fiber is illuminated with the beam of interest, rather than focusing the beam on a specific core. This type of excitation is similar to that used in the previous works, allowing the results of this study to be placed in direct context of the previous works. Note that this comparison is addressed in our response to the Reviewer's fourth comment (see R1.4).

After carefully reviewing Fig. 1, we understand the Reviewer's concern, as the figure does seem to suggest that each core is illuminated by a separate beam. To clarify, we have revised Fig. 1 to show an incident light field that fully covers the nanostructured end face of the multicore fiber and includes two beams with different angles of incidence that are coupled into two selected cores, highlighted in green and magenta. Please see the updated figure and figure caption in the response letter (Fig. RL1).

Fig. RL1 [Fig. 1 of the manuscript]: Illustration of the concept of all-fiber integrated angular demultiplexing using nanoprinted periodic structures on single-mode multicore fibers (left: input side, right: output side). Note that the **two colored arrows refer to an incident light field that includes two beams with different angles of incidence that are coupled into two selected cores**. The lower right inset shows an example image of a functionalized multicore fiber that contains a nanoprinted periodic structure on each of the cores (the **white** scale bar corresponds to a length of 50 μ m).

To respond to the Reviewer's point, we have modified the text of the Methods section: ... Optical measurements were performed by illuminating the nanostructured end of the sample with collimated light ($\lambda_0 = 1550$ nm) and capturing the output pattern with a camera (c.f., Supplementary Note 6). **Note that similar to previous works [link], [link], the entire fiber end face is illuminated.** To compensate for the limited dynamic range of the camera, ...

R1.3: The authors mentioned that “To eliminate the power dependence on the intensity of the incident light, all data discussed in this work is normalized to the output power of a bare fiber that contains no nanostructure at its end face at normal incidence”. Why does the incident light in the experiment exhibit varying intensities? It would be beneficial to provide the normal incidence efficiency $P_b(\theta=0)$ for readers to understand the potential efficiency achievable with multicore fibers.

We thank the Reviewer for the comments, which we address individually below.

Discussion of normalization: The sentence in question essentially refers to the normalization of nanostructure-enhanced coupling efficiency. After careful re-reading, we agree that the sentence was awkwardly worded and potentially misleading. Our intended meaning is that the light coupling efficiency at normal incidence varies for each grating because each structure removes a specific fraction of power from the incident beam. Note that in general, the fraction of power coupled into each nanostructure-enhanced core is less than that of a bare fiber due to the additional reflection and scattering introduced by the nanostructure itself (for details see our response to the seventh comments of Reviewer #2 (R2.7)). To account for this, all power measurements have been normalized to the output power of a bare fiber under normal incidence, which is the maximum achievable value of the coupling efficiency.

To clarify this point, we have revised the following parts of the main text:

... The most important benchmark parameter that determines the effectiveness of the coupling at a given angle of incidence is the coupling efficiency. This parameter describes the efficiency of converting the power of a plane wave incident on a fiber at the angle θ into one of the supported guided modes. From a practical point of perspective, the output power $P(\theta)$ is measured for different incident angles. **As each structure removes a specific fraction of power from the incident beam**, all data discussed in this work is normalized to the output power of a bare fiber that contains no nanostructure at its end face at normal incidence $P_b(\theta = 0^\circ)$, leading to: ...

... In general, $P_b(\theta = 0^\circ)$ results in the highest possible coupling efficiency ($\eta = 1$) as ...

We thank the Reviewer for the comment regarding the efficiency of the bare fiber. In response, we have added this value to the manuscript (section 4.2, a paragraph “*Data analysis*”) as follows: ... For comparison, a bare MCF characterized in the same setup yields **37.4 μ W of total output power** from all cores combined under normal incidence. ...

R1.4: “Compared to reported literature [7, 11, 22, 23], the achieved efficiencies of several percent are remarkably high and are mainly due to the discussed optimization procedure.” It is recommended to include a comparison table to provide a comprehensive and fair assessment of the enhancements facilitated by their method.

We thank the Reviewer for highlighting the importance of comparison with the literature. In response, we have addressed this point by comparing our results with those reported in previous studies, as shown in the following table (Tab. RL1), which we have included in the

Supplementary Information. This addition to the Supplementary Information allows us to keep the main text focused and concise, highlighting the novel findings and advances of our study without introducing extensive comparisons that could disrupt the flow of the primary discussion. By placing the table in the Supplementary Information, readers interested in a more detailed comparison can access it without being distracted from the main narrative of the manuscript.

To address the Reviewer's comment, the main text of the manuscript has been changed as follows: ... An exact overlap of the local maxima of the coupling efficiency with the target angles can be recognized, showcasing the relevance of the optimization procedure. Compared to reported [REFs], the achieved efficiencies of several percent are remarkably high and are mainly due to the discussed optimization procedure (**a comparison of the various published results with those obtained in this work is presented in the Supplementary Note 8**). ...

The new section of the Supplementary information reads as follows: ...

Supplementary Note 8. Comparison of fiber-based devices performance and fabrication methods

Supplementary Table 5 shows a comparison of the device presented in this work with devices from previous studies in terms of nanostructure design, fabrication methods, fiber types, and mode characteristics. Previous studies have solely focused on implementing various configurations of nanostructures on single-core fibers, such as plasmonic nanodot gratings [link], SiN annular gratings [link], both fabricated by modified electron beam lithography, and complex periodic ring nanostructures by direct laser writing [link], [link]. In contrast, this study is the first to fabricate annular gratings directly on a custom-designed multicore fiber, integrating seven single-mode cores without modal crosstalk.

Previous studies have reported much lower coupling efficiencies at an incident angle of 40°. For example, the plasmonic nanodot grating by N. Wang et al. [link] achieved only 0.02%, while the SiN ring grating by O. Yermakov et al. [link] reached 0.006%. In comparison, the device in this study achieved a coupling efficiency of 1.4% to 1.5%, a significant improvement over previous designs for a single-mode fiber design, demonstrating improved collection capabilities for off-axis light. Notably, while the multimode fiber system listed in the table achieved even higher coupling efficiencies (up to 6.1%) [link], the larger core and multimodeness do not allow angular multiplexing and may have practical disadvantages compared to the single-mode fibers used in this work. A similar conclusion can be drawn by comparing the values at an incident angle of 70°.

This comparison highlights the technical advancement achieved in this study, where the multi-core fiber design and optimized nanostructures enable superior light coupling efficiency, particularly at high angles of incident, outperforming previous single-core fiber designs.

Tab. RL1 (Supplementary Table 5): Comparison of the key performance

indicators of the presented device with those reported in the literature.

	Wang et al.	Yermakov et al.	Yermakov et al.	Zeisberger et al.	this work
nanostructure	plasmonic nanodot grating	SiN annular grating	polymer annular grating	polymer annular grating	polymer annular grating
lithographic method	modified e-beam	modified e-beam	direct laser writing	direct laser writing	direct laser writing
fiber	SMF-28	SMF-28	SMF-28	Optran Ultra WFGE	in-house multicore fiber
mode characteristics	single-mode	single-mode	single-mode	multimode	single-mode
No. cores	1	1	1	1	7
single-core / total coupling efficiency (at 40°)	0.02%	0.006%	1.8% (single-pitch structure #1) 0.06% (aperiodic structure)	6.1%	1.4% / 1.5%
single-core / total coupling efficiency (at 70°)	0.002%	0.2%	0.02% (single-pitch structure #3) 0.03% (aperiodic structure)	0.6%	0.25% / 0.4%
capability for angle demultiplexing	no	no	no	no	yes

R1.5: There is a figure reference error with corrupted characters in Section 4.1.

We thank the Reviewer for highlighting the corrupted characters in Sec. 4.1, which have been corrected in the revised version of the manuscript: ... Applying the equations from steps 1 to 3 ..., with the corresponding angular distribution of the coupling efficiency shown by the green curves in Figs. **S4g-S4h**. ... Subsequent numerical optimization (step 4) produces the final optimized parameters ..., showing an exact overlap of the maximum coupling efficiency with the target angle of 40° (blue curves in Figs. **S4g-S4h**).

Response to comments of Reviewer #2

The present manuscript presents an all-fiber integrated angular demultiplexer by printing periodic structures on a multicore fiber tailored to couple light at specific angles. The present work shows an alternative of a microstructure on the tip of an optical fiber that allows to selectively couple light with different angle of incidence into different fiber cores. The work presents an integrated solution to capture or emit light from angles where usually conventional optical fibers are limited. The impact of this work seems mostly for free-space applications such as communications, spectroscopy, or integrated photonics, for just mentioning the most relevant ones. Although structures are simple and design is not very complicated, it does require a very specialized lithography system; nonetheless, the progress in this area should simplify the fabrication of such structures. The manuscript is well written and the structure of the article is appropriate. I recommend the article for publishing but after some aspects are addressed.

We sincerely thank the Reviewer for the overall positive assessment of our work, the recommendation for publishing it in Nature Communications and for the thorough review of the manuscript. Below, we have provided detailed responses to each of the Reviewer's comments. We hope that our revisions have sufficiently addressed all concerns and contributed to a stronger and more refined version of the manuscript.

R2.1: A major aspect missing is how the structure works for a light source with a light source of broadband spectrum. The only example of this can be found in Fig. 3b, but no further comments or analysis are provided. For instance, calculations of how much the coupling efficiency changes for the structure within a short wavelength range, such as 5 or 10 nm wavelength span. Another example would be to extend the analysis of coupling efficiency for the C-band, since in the application it is mentioned that it can potentially be used for optical communications.

We thank the Reviewer for highlighting the important issue of the spectral dependence of the device in the context of wavelength variations of the incident beam at telecom wavelengths. To address this, we simulated the effect of spectral variations on both the diffraction angle and the collection efficiency within the telecom C-band. Specifically, the coupling efficiency was calculated for the nanostructure-enhanced MCF that has a design angle of $\theta_{-1} = 40^\circ$ (grating #3 in Tab. 1) in the vicinity of this angle for various wavelengths between 1530 nm and 1565 nm (Fig. RL2(a)). The results show a minor change in the characteristics, with a negligible variation in the amplitude of the coupling efficiency and a slight spectral shift in the angle at which the coupling efficiency peaks (maximum coupling angle θ_{max}). To quantify this dependence, the maximum angle was plotted as a function of wavelength (Fig. RL3(b)), showing a linear relationship with a slope of $\Delta\theta_{max}/\Delta\lambda = 0.02^\circ/nm$, which is practically insignificant.

Fig. RL2 [Supplementary Figure 9]: Simulated impact of the wavelength dependence on the coupling efficiency. (a) Angular dependence of the coupling efficiency for the grating with a design angle of $\theta_{-1} = 40^\circ$ (grating #3 from Tab. 1) for different wavelengths within the telecom C-band (indicated by different colors). (b) Angle of maximum coupling efficiency as a function of wavelength within the C-band. The resulting slope of this linear dependence is $\Delta\theta_{max}/\Delta\lambda = 0.02^\circ/\text{nm}$.

To address the Reviewer's comments, the above discussion has been added to the Supplementary Information

Supplementary Note 5. Selectivity and spectral dependence of fiber-based device

Spectral dependence. We analyze the effect of spectral variations on both the diffraction angle and the collection efficiency within the telecom C-band (from 1530 to 1565 nm). Specifically, the coupling efficiency was calculated for the nanostructure-enhanced multicore fiber that has a design angle of $\theta_{-1} = 40^\circ$ at 1550 nm (grating #3 in Tab. 1) in the vicinity of this angle for various wavelengths between 1530 nm and 1565 nm (Supplementary Figure 9(a)). The results show a minor change in the characteristics, with a negligible variation in the amplitude of the coupling efficiency and a slight spectral shift in the maximum coupling angle θ_{max} . To quantify this dependence, the maximum angle was plotted as a function of wavelength (Supplementary Figure 9(b)), showing a linear relationship with a slope of $\Delta\theta_{max}/\Delta\lambda = 0.02^\circ/\text{nm}$.

and the main text has been revised as follows: ... Moreover, to demonstrate the spectral dependence of the device, the angular dependence of the coupling efficiency was simulated for the nanostructure-enhanced fiber with a design angle of $\theta_{-1} = 40^\circ$ (grating #3 in Tab. 1) within the telecom C-band. The results show negligible variation in the coupling efficiency amplitude and minimal spectral shift in the maximum coupling angle of $0.02^\circ/\text{nm}$, which is practically insignificant (details can be found in the Supplementary Note 5). ...

R2.2: In Fig. 6 the distribution of the periodic structures for the different angles is presented. How were the different structures corresponding for different incident angles assigned? Is

there an optimal distribution or some guidelines for this? This is a major aspect since the output patterns of the MCF for light launched from different angles does couple to different cores, especially at small angles. How can this be minimized and please provide some numbers of “crosstalk” for the proposed device. For example, for an unknown direction light source (1550 nm), what is the relationship for the intensity output at different cores and how the incidence angle can be determined? Additional discussion on the selectivity of the device must be included.

We thank the Reviewer for the important comments, which are addressed below. Please note that we have divided the questions into two thematically related groups.

A. Reviewer comment: *“How were the different structures corresponding for different incident angles assigned? Is there an optimal distribution or some guidelines for this?”*

The design procedure for correlating a selected coupling angle to the mode of a specific core using the periodic nanostructure follows the steps outlined in Section 2.4 of the manuscript and Supplementary Notes 2 and 3 of the Supplementary Information. It involves four main steps: (i) determining the pitch based on the lattice resonance condition, (ii) deriving the position of the first ring using a specific Bessel function equation, (iii) determining the geometric parameters of the rings, and (iv) performing numerical optimization. For further details, we kindly refer the Reviewer to these sections in the main text and Supplementary Information. Based on the mentioned discussions we believe that the procedure for determining the geometric parameters of the periodic structures is sufficiently explained already.

In response to the Reviewer's comment, we have revised the text to more clearly emphasize the use of this particular design approach in the following sections of the main text:

... The device is based on algorithmically optimized nanostructures **using a 4-step procedure and** fabricated by 3D nanoprinting on the facets of multicore fibers with seven SMF-28-type cores ...

... The study includes detailed theoretical analysis, **introduces a novel computational optimization approach that is based on a four-step procedure**, experimental implementation via fiber drawing and 3D nanoprinting, and detailed optical characterization, which together highlight the advantages and limitations of the concept. ...

B. Reviewer comment: *“This is a major aspect since the output patterns of the MCF for light launched from different angles does couple to different cores, especially at small angles. How can this be minimized and please provide some numbers of “crosstalk” for the proposed device. For example, for an unknown direction light source (1550 nm), what is the relationship for the intensity output at different cores and how the incidence angle can be determined? Additional discussion on the selectivity of the device must be included.”*

Small angle regime: The incoupling of light at small angles of incidence effectively results from the incoupling of the 0th order diffraction. The appearance of a 0th order is an intrinsic property of a diffraction grating and defines the operating range of the device to angles greater than 12°. Strategies to reduce the impact of the 0th order include computationally

optimized gratings to transfer more power into a selected diffraction order (e.g., Dammann gratings [link]) or complex metasurface structures [link]. This aspect has already been mentioned in the Discussion and Conclusion section of the manuscript: ... The concept presented is based on axially symmetric diffraction gratings designed to couple light into a specific core within a defined angular interval. One direction for future optimization is to explore alternative grating types. For example, Dammann gratings modified by algorithmic optimization have been shown to effectively suppress the zeroth order of diffraction while selectively directing light output to higher diffraction orders [REF]. In addition, complex three-dimensional structures such as photonic crystal-based superprisms [REF] could potentially be applied directly to fibers using 3D nanoprinting for redirecting the light. Another topical approach is the use of metasurfaces or phase holograms, which allow precise phase manipulation to control [REF]. ...

We thank the Reviewer for highlighting the important point about crosstalk, which is a critical issue in the context of multicore fibers, as cross-coupling between modes of adjacent cores can lead to unwanted power transfer between cores. This concern is also critical to the concept discussed here, as power exchange between cores must be prevented to ensure unambiguous correlation between the angle of incidence and the power in each respective core.

To quantify the crosstalk of the MCF used, additional experiments were performed. Specifically, to evaluate the modal coupling between the cores (i.e., intercore cross-coupling), broadband light was injected into the central core of the MCF (using a combination of a supercontinuum light source, coupling optics, and spectral diagnostics) and the power transmission of the central core and one of the side cores was measured. This procedure was performed for three bending radii by placing a portion of the MCF in a support plate with customized grooves to create precise bending conditions (c.f. our response to the fifth comment of the Reviewer (R2.5)). Note that the related experimental setup included microscopes at the input and output sides of the MCF to (i) directly observe which core was excited and (ii) collect light from a specific core at the output. To remove the spectral characteristics of the light source and optics, the difference of the power transmission (on a logarithmic dB scale) between the center-center and center-side core configurations $\Delta t_{dB} = t_{cs,dB} - t_{cc,dB}$ was calculated (Fig. RL3(a)).

The results show no inter-core cross-coupling up to an extinction ratio of 25 dB over the entire spectral range and especially at the operating wavelength of $\lambda_0 = 1.55 \mu\text{m}$ (vertical gray dashed line in Fig. RL3), which is important for unambiguously correlating angles of incidence and power in the corresponding core. It should be noted that the current dynamic range of this setup is currently limited to about 25 dB, probably due to residual light of the center core entering the objective when the output objective is adjusted to measure one of the side cores.

To further confirm the absence of intermodal cross-coupling at the operating wavelength, narrowband light (center wavelength 1550 nm, bandwidth 10 nm) was coupled into the center core and the output mode was imaged with an infrared camera (details in the caption of Fig. RL3). To increase the dynamic range of the camera, grey scale images were taken at different exposure times (1 ms to 200 ms), increasing the initial 8-bit dynamic range of a single image from 23 dB to 46 dB (Fig. RL3(b)-(e)). This procedure results in strong saturation in the middle of the images dominated by light from the central core at high exposure times (Fig. RL3(d)-(e)), while none of the images show any sign of intensity in the side cores, confirming the absence of intermodal cross-coupling. Note that the feature on the

right of (d) and (e) is a confirmed imaging artifact caused by reflections). A more accurate assessment of the cross-coupling behavior will be pursued in a future study using improved experimental approaches such as butt-coupling and artifact-free imaging as part of a dedicated study on multicore fiber.

Fig. RL3 [Supplementary Figure 4]: (a) Spectral distribution of the transmission difference between the center-center and center-side configurations (details in the main text) for three different bend radii (red: 2 cm, green: 4 cm, blue: 6 cm). The vertical gray dashed line refers to the operating wavelength of 1.55 μm . The images on the right (b)-(e) show the measured output intensity distribution when light is injected into the center core at different exposure times of the IR camera used (indicated in the lower right of each image, taken with an ABS-Jena IK1513 camera). A bandpass filter (FWHM bandwidth: 10 nm, central transmission wavelength: 1550 nm) was inserted into the input beam path to selectively measure the properties at the operation wavelength (indicated by the vertical gray dashed line in (a)). The feature to the right of (d) and (e) is a confirmed imaging artifact caused by reflections within the imaging setup, while the visible deformation of the center beam in (d) and (e) is due to imaging artefacts.

To account for the Reviewer’s comment, we have included this part into the Supplementary Note 1 as

Intermodal cross-coupling:

To quantify the crosstalk of the MCF used, additional experiments were performed. Specifically, to evaluate the modal coupling between the cores (i.e., intercore cross-coupling), broadband light was injected into the central core of the MCF (using a combination of a supercontinuum light source, coupling optics, and spectral diagnostics) and the power transmission of the central core and one of the side cores was measured. This procedure was performed for three bending radii by placing a portion of the MCF in a support plate with customized grooves to create precise bending conditions. Note that the related experimental setup included microscopes at the input and output sides of the MCF to (i) directly observe which core was excited and (ii) collect light from a specific core at the output. To remove the spectral characteristics of the light source and optics, the difference of the power transmission (on a logarithmic dB scale) between the center-center and center-side core configurations ($\Delta t_{dB} = t_{cs,dB} - t_{cc,dB}$) was calculated (Supplementary Fig. 4(a)). The

results show no inter-core cross-coupling up to an extinction ratio of 25 dB over the entire spectral range and especially at the operating wavelength of $\lambda_0 = 1.55 \mu\text{m}$ (vertical gray dashed line in Supplementary Fig. 4(a)), which is important for unambiguously correlating angles of incidence and power in the corresponding core. It should be noted that the current dynamic range of this setup is currently limited to about 25 dB, probably due to residual light of the center core entering the objective when the output objective is adjusted to measure one of the side cores. To further confirm the absence of intermodal cross-coupling at the operating wavelength, narrowband light (center wavelength 1550 nm, bandwidth 10 nm) was coupled into the center core and the output mode was imaged with an infrared camera (details in the caption of Supplementary Fig. 4). To increase the dynamic range of the camera, grey scale images were taken at different exposure times (from 1 ms to 200 ms), increasing the initial 8-bit dynamic range of a single image from 23 dB to 46 dB (Supplementary Figs. 4(b)-4(e)). This procedure results in strong saturation in the middle of the images dominated by light from the central core at high exposure times (Supplementary Figs. 4(d)-4(e)), while none of the images show any sign of intensity in the side cores, confirming the absence of intermodal cross-coupling.

and modified the main text of the manuscript as follows: ... At the operation wavelength, the modes have losses of the order of 1 dB/km (Supplementary Figure 3). **To quantify intermodal crosstalk, broadband light was injected into the center core and power transmission was measured for both the center and side cores at three bend radii, showing no crosstalk between the cores down to an extinction ratio of 25 dB (details can found in the Supplementary Note 1, Supplementary Fig. 4). This absence of intermodal cross-coupling was additionally confirmed by coupling narrowband light (1550 nm, 10 nm bandwidth) into the center core and imaging the output with an infrared camera at different exposure times (from 1 ms to 200 ms, achieved dynamic range 46 dB), showing no evidence of intermodal cross-coupling, thus meeting the requirements of this application. ...**

Selectivity of the device: We thank the Reviewer for the valuable comment on the selectivity of the device. We address this here by explaining how the device can identify an unknown direction of a light source, a common application scenario at near-infrared wavelengths. The concept is based on comparing the power levels in different cores, since the power in a single core alone does not provide sufficient information to determine the angle of incidence. In detail, the procedure includes (i) identifying the core with the highest power, which corresponds to one of the colored curves in Fig. 3(a); and (ii) comparing the power levels of modes of several related cores in order to unambiguously determine the angle of incidence. This approach is exemplified here by examining the simulated coupling efficiencies for three closely spaced angles around an incident angle of 40° ($\theta = 38^\circ, 40^\circ, 42^\circ$). The corresponding values of the coupling efficiencies of the gratings with the design angles $30^\circ, 40^\circ$ and 50° – $\eta_{30^\circ}, \eta_{40^\circ}$ and η_{50° – are given in Tab. RL2. Additionally, the ratios between these coupling efficiencies ($\eta_{30^\circ} / \eta_{40^\circ}$ and $\eta_{50^\circ} / \eta_{40^\circ}$) are also given. The values of the coupling efficiencies for the three angles of incidence ($\theta = 38^\circ, 40^\circ, 42^\circ$) are significantly different, demonstrating that the concept allows unambiguous identification of the angle of incidence, in the current example with a selectivity of approximately 2° .

Note that the procedure is simplified if the total power of the incident beam on the fiber

surface is known, as this allows the absolute fraction of power coupled into the cores to be determined.

Tab. RL2 [Supplementary Table 2]: Example of selected simulated coupling efficiencies, taken from Fig. 3(a), for three closely spaced angles of incidence ($\theta = 38^\circ, 40^\circ, 42^\circ$), showing the selectivity of the device. The first three rows show the coupling efficiencies for three different cores (i.e., nanostructures) resulting from the curves shown in Fig. 3(a) (η_{30° : blue curve, η_{40° : cyan curve, η_{50° : green curve). The two lower rows show the corresponding ratios of the coupling coefficients (top: $\eta_{30^\circ}/\eta_{40^\circ}$, bottom: $\eta_{50^\circ}/\eta_{40^\circ}$).

θ	38°	40°	42°
η_{30° [%]	0.0053	0.0282	0.0407
η_{40° [%]	2.2168	2.9151	2.1446
η_{50° [%]	0.0767	0.0433	0.0031
$\eta_{30^\circ} / \eta_{40^\circ}$	0.0024	0.0097	0.0190
$\eta_{50^\circ} / \eta_{40^\circ}$	0.0346	0.0149	0.0015

To address the Reviewer's comment, we have included this example in the Supplementary Information

Supplementary Note 5. Selectivity and spectral dependence of fiber-based device

Selectivity. To prove the selectivity of the developed fiber-based device, we derive the procedure including (i) identifying the core with the highest power, which corresponds to one of the colored curves in Fig. 3(a); and (ii) comparing the power levels of modes of several related cores in order to unambiguously determine the angle of incidence. We examine the simulated coupling efficiencies at $\lambda_0 = 1.55 \mu\text{m}$ for three closely spaced angles around an incident angle of 40° ($\theta = 38^\circ, 40^\circ, 42^\circ$). The corresponding values of the coupling efficiencies of the gratings with the design angles $30^\circ, 40^\circ$ and 50° – $\eta_{30^\circ}, \eta_{40^\circ}$ and η_{50° – are given in Supplementary Tab. 2 as well as the ratios between these coupling efficiencies ($\eta_{30^\circ}/\eta_{40^\circ}$ and $\eta_{50^\circ}/\eta_{40^\circ}$). One can notice that the values of the coupling efficiencies for the three angles of incidence are significantly different and well separated, demonstrating that the concept allows unambiguous identification of the angle of incidence. ...

and modified the main text as follows: ... This effect can be clearly seen in the supplementary video (c.f. Video~1), which shows the dynamic change of the power distribution across the different cores in case the angle of incident is scanned. ~~In summary, the fiber-integrated angle demultiplexer presented here allows analyzing an input angle spectrum in the domain $\theta > 15^\circ$.~~

A potential application scenario that often arises in the context of near-infrared light sources is the identification of an unknown direction of a light source. It is important to note that such a determination using the fiber-based device discussed requires a

comparison of the power levels in different cores, as a single-core measurement does not provide sufficient information. Specifically, the procedure involves (i) identifying the core with the highest output power, which corresponds to one of the colored curves in Fig. 3(a), and (ii) comparing the power levels of modes from several related cores, which allows the angle of incidence to be uniquely determined. An example of this procedure is shown in the Supplementary Note 5. ...

R2.3: Since the structures are symmetrical, polarization dependence is not expected. However, this is not mentioned. How does the polarization of light interacting with the structure play a role?

We thank the Reviewer for this comment. The on-fiber nanostructures presented in this study show a slight dependence on the polarization state of the incident light due to the inherent polarization sensitivity of gratings in general. Note that the polarization dependence of nanostructure-enhanced light coupling has already been investigated in one of our previous papers [link]. Specifically, Fig. 4 of that paper compares experimental and simulated angular distributions of the coupling efficiency of SiN gratings on single-core fibers in the case the incident beam has either a TE or a TM polarization. The results show minimal differences for the two input polarization states, especially within the angular ranges of the local peaks, indicating that polarization dependence is a relatively minor effect for the concept investigated here.

To account for the Reviewer's comment, we have amended the main text as follows: ... This structure acts as an axially symmetric diffraction grating with a -1^{st} diffraction order corresponding to the specific angle $\theta_m = \theta_{-1}$ [m : diffraction order, Fig. 2(d)]. Therefore, by creating gratings with different optimized coupling angles on the different cores of an MCF, the incoming angular spectrum can then be determined by analyzing the power in the different cores at the fiber output side. **Note that, as shown in a previous study [link], the effect of the polarization of the incident light is practically negligible, especially within the angular ranges of the local peaks. ...**

R2.4: In Table 1 of the supplementary material it can be observed that the error in the heights of the periodic structures is big while the designed and optimized structure are compared. Why is this? In addition, the heights of these structures need fabrication resolutions of 10 nm. What is the resolution of the fabrication process in height and what are the penalties in the response of the structures if a resolution of 0.1 μm is only achievable?

We thank the Reviewer for this comment. Below, we will first address the issue of the difference between the analytical and optimized heights, and then discuss the effect of height variation on optical performance.

A. Height difference between analytical and optimized designs

The difference in element heights between the analytical and optimized designs is due to the following considerations: In the analytical design, all elements are assigned the same height, determined by the condition that under normal incidence, the phase difference between the wave passing through the polymer and the air domains is equal to π , calculated as $h = \lambda_0 / (2 \Delta n)$. This is part of the optimization process described in Step 3, Section S2 of the Supplementary Information. However, the height of the final numerically optimized structures

may differ due to (i) scattering of incident light at the polymer elements for large incident angles, which can lead to partially shielding, and (ii) different phase retardation within the polymer elements for non-normal incidence. It is important to note that the identical height used in the analytical steps serves as a starting point for numerical optimization.

We have changed the text of the Methods section to emphasize this point: ... where $\Delta n = n_p - n_{\text{air}}$ is the difference between the refractive indices of polymer and air. **Note that the analytically determined heights result from the condition that under normal incidence, the phase difference between the wave passing through the polymer and the air domains is equal to π , which implies that all elements have the same height in this step. ...**

At this point, we would like to mention that the term "error" used to describe the difference in height between the analytic and the optimization design may be misleading because it is not an error resulting from any type of inaccuracy. To address this issue, we have replaced the term "relative error" with "relative difference" throughout the Supplementary Information (main text and Tab. 1). For example, the following sections now read as follows:

... Using the same procedure, we obtain optimized designs for the different local maxima angles. A full comparison between the analytic-based and optimized designs with the corresponding relative ~~errors~~ **differences** is given in Tab. 3. The relative ~~error~~ **difference** for different parameters calculated here was defined as ...

... Please note that for the $\delta_{\eta, \text{may}}$ the absolute value operator was eliminated, so the sign minus in the relative ~~error~~ **difference** means the higher values of the in-coupling efficiencies for the analytic-based design. ...

B. Impact of fabrication-induced inaccuracies in achieving the optimized height

Another important aspect is to uncover the effect of variations in height from the optimized value. To address this point, we performed additional simulations that determine the angular distribution of the coupling efficiency for a selected configuration (grating #3 in Table 1, $\theta_{-1} = 40^\circ$) for different heights of the grating elements (Fig. RL4). As can be seen, unlike pitch, ring width, and position of the first ring, variations in height have little effect on the performance of the device. In fact, there is practically no change in the angular position of the maximum, with a slight change in the amplitude of the coupling efficiency. Specifically, it can be seen that the amplitude of the coupling efficiency changes by less than 0.2% for a decrease or increase in height of 300 nm, which is less than 10% of the maximum value.

Fig. RL4 [Supplementary Figure 8]: (a) Angular dependence of the coupling efficiency for a nanostructure-enhanced single-mode fiber with a design angle of incidence of $\theta_{-1} = 40^\circ$ (grating #3 from Tab. 1) for varying grating element heights. (b) Local maximum of coupling efficiency as a function of element height.

These simulations clearly demonstrate that the required vertical resolution in 3D nanoprinting can be significantly coarser than 10 nm. The accuracy of our nanoprinting device, approximately 50 nm, is thus sufficient for the operational requirements of the device presented in this study. Note that this value is determined by the accuracy of the piezo stage used to position the sample vertically during 3D nanoprinting.

To address the Reviewer's comments, we have included this discussion in the Supplementary Information

... **Supplementary Figure 8 shows the angular dependence of the in-coupling efficiency in the vicinity of local maximum for different heights of the grating #3 from Tab. 1 ($\theta_{-1} = 40^\circ$). In fact, there is practically no change in the angular position of the maximum, with a slight change in the amplitude of the coupling efficiency by less than 0.2% for a decrease or increase in height of 300 nm.**

and expanded the main text of the manuscript as follows: ... Note that the numerically optimized height of the grating elements oscillates around the design height parameter $h = \lambda_0 / (2\Delta n)$ defined in step 3 (see Supplementary Note 4). **Note that, as shown by additional simulations (see Supplementary Figure 8), variations in the height of the grating elements have minimal effect on the performance of the device, with negligible changes in the angular position and amplitude of the coupling efficiency when realistic height variations are assumed (vertical position precision 50 nm).** ...

R2.5: In the supplementary material, section S1, the characterization of the MCF is explained. The cores of the MCF were designed to not present coupling among the cores. Although bending losses are nicely characterized, how the bending affects the mode coupling among cores is not shown in this study. This is important because the fiber design is part of this manuscript. An explanation of this aspect and some calculations should be included.

We thank the Reviewer for the comment on bending-induced coupling loss. This point is

addressed in our response to the Reviewer's second comment, and we kindly refer the Reviewer to R2.2 and the corresponding new section in the Supplementary Information.

R2.6: It contains detailed information regarding the design of the periodic structures. More detailed information is provided in the supplementary material. However, during the reading of the main article sometimes it is not clear that all the design and characterization is performed at a wavelength of 1550 nm. This must be mentioned in all the plots and tables.

We thank the Reviewer for this comment, which we have addressed by making the following changes to the various captions.

Caption of Tab. 1: ... Geometric parameters of the seven optimized designs of axially symmetric polymer structures resulting from the optimization for the different target angles ($\lambda_0 = 1.55 \mu\text{m}$). ...

Caption of Fig. 3: ... Results of the optimization of seven axially symmetric nanogratings to excite the fundamental mode in the individual cores of the MCF at different incident angles ($\lambda_0 = 1.55 \mu\text{m}$). ...

Caption of Fig. 5: ... All curves are normalized to the coupling efficiency of a single bare core at normal incidence (Eq. 1, $\theta = 0^\circ$, $\lambda_0 = 1.55 \mu\text{m}$) ...

Caption of Fig. 6: ... Selected measured images of the output intensity of the nanostructured-enhanced multicore fiber (Video 1), showing the power distribution across the different cores at different angles of incidence (given at the bottom of each image, $\lambda_0 = 1.55 \mu\text{m}$). ...

Caption of Supplementary Fig. 5: ... Details of the optimization procedure, demonstrated here on the example of $\theta_0 = 40^\circ$ (described in all plots by the vertical black dashed line, $\lambda_0 = 1.55 \mu\text{m}$). ...

Caption of Supplementary Tab. 1: ... Comparison between the analytic-based (steps 1--3) and numerically optimized nanostructure designs ($\lambda_0 = 1.55 \mu\text{m}$). ...

Caption of Supplementary Fig. 6: ... Angular distribution of coupling efficiencies for the seven designs of axially symmetric structures with different local maxima ($20^\circ < \theta_m < 80^\circ$, in steps of $\Delta\theta = 10^\circ$, shown by the vertical gray dashed lines, $\lambda_0 = 1.55 \mu\text{m}$) ...

Caption of Supplementary Fig. 12: ... Normalized intensity (in %) distributions of the transmission grating as a function of diffraction angle for four different periods (i.e., values of a pitch, $\lambda_0 = 1.55 \mu\text{m}$): ...

R2.7: As the periodic structures are fabricated on top of each SM core and are designed to capture light at different incidence angle, I missed how the losses are due to the periodic structures while the incidence is normal. Please clarify this.

We thank the Reviewer for the comment regarding the coupling efficiency of the different

structures in the case of normal incidence. To address this comment, we have summarized the simulation and experimental values in the following table (Tab. RL3), which shows the change in coupling efficiency in the presence of the polymeric nanostructures. Here the nanoprinted structures reduce the coupling efficiency at normal incidence, indicating that the device is designed to operate predominantly in the range of non-normal incidence. Note that the reflectivity of a plane wave from a bare single-mode fiber under normal incidence is about 4%.

Tab. RL3 [Supplementary Table 3]: Summary of simulated and measured coupling efficiency of the various geometries for normal incidence (details of corresponding geometric parameters can be found in Tab. 1 of the main text).

fiber-based devices	$\eta_{\text{sim}}(\theta = 0^\circ)$	$\eta_{\text{exp}}(\theta = 0^\circ)$
bare fiber	1	1
nanostructure #1 ($\theta_{-1} = 20^\circ$)	0.258	0.305
nanostructure #2 ($\theta_{-1} = 30^\circ$)	0.080	0.118
nanostructure #3 ($\theta_{-1} = 40^\circ$)	0.044	0.143
nanostructure #4 ($\theta_{-1} = 50^\circ$)	0.097	0.197
nanostructure #5 ($\theta_{-1} = 60^\circ$)	0.257	0.360
nanostructure #6 ($\theta_{-1} = 70^\circ$)	0.380	0.446
nanostructure #7 ($\theta_{-1} = 80^\circ$)	0.390	0.424

To address the Reviewer's comment, we have added the above discussion to the Supplementary Information

... **The nanoprinted polymer structures reduce the coupling efficiency at normal incidence, indicating that the device is designed to operate predominantly in the range of non-normal incidence (Supplementary Tab. 3).**

and changed the text of the manuscript as follows: ... Note that the light collection efficiency of the functionalized MCF exceeds that of its unstructured counterpart by 2 ($\theta > 16^\circ$) to 4 ($\theta > 30^\circ$) orders of magnitude. Both experimental and numerical results are in good agreement, especially for $\theta < 60^\circ$. **The nanoprinted polymer structures reduce the coupling efficiency at normal incidence, indicating that the device is designed to operate predominantly in the non-normal incidence range (Supplementary Table 3 shows the coupling efficiency of all configurations).** ...

R2.8: The structures are demonstrated on a MCF, but this type of application should work as well on all-fiber spatial multiplexers. Please include this in the discussion of the manuscript.

We thank the Reviewer for this insightful comment, as spatial multiplexing is indeed a promising additional application for the concept demonstrated in this study. The principles

underlying angle-sensitive coupling and mode excitation can be readily applied to spatial multiplexing systems, where they could enhance the distribution of light across multiple channels. This extension could open new opportunities for advanced light manipulation, routing, and signal processing in various fiber-based applications.

To account for the Reviewer's comment, we have extended the Discussion section as follows: ... The presented approach to collect and analyze light as well as the combination of multicore fibers and fiber-integrated nanostructures as a whole are promising for a wide range of applications including wavelength-division-multiplexing in high-capacity telecommunication networks, trace gas detection in environmental sciences, molecular sensing in bioanalytics, and efficient light coupling within integrated photonics and nano-optics. **The concept is particularly well suited for fiber-based spatial multiplexing systems, where the angular power spectrum can be distributed over multiple spatial channels for efficient light collection and routing, and can particularly benefit from highly dispersive nanostructures such as metasurfaces [link] or photonic crystal-based superprisms [link].** It is important to highlight that the fabrication process employed generally allows for the creation of structures with significantly greater complexity, thereby substantially broadening the spectrum of potential applications. ...

R2.9: References cited in the manuscript are adequate but some other works with similar structures should be included and commented:

<https://pubs.acs.org/doi/10.1021/acs.nanolett.0c04463>

<https://doi.org/10.1038/s41467-024-52256-y>

<https://doi.org/10.1038/s41598-017-04490-2>

<https://doi.org/10.1364/PRJ.521005>

<https://doi.org/10.1038/s41467-022-31902-3>

<https://doi.org/10.1038/s41377-023-01222-2>

We thank the Reviewer for the comment about including additional literature, which we have addressed below by grouping the publications mentioned into different categories and discussing their relevance in the context of our work.

A. Studies that address fiber interfacing: The following works target various types of on-fiber nano- and microstructures used for light focussing.

W. Hadibrata et al [link]: In this paper, the authors present an on-fiber inverse design nanostructure for strong light focusing. Similar to our approach, the grating structures consist of concentric ring-like arrays fabricated using 3D nanoprinting for fiber interfacing. This study is consistent with several of our previous works on fiber-interfaced holographic focus generation. It was already included in the submitted version of the manuscript.

R. S. R. Ribeiro et al [link]: In this paper, the authors present a fiber interfaced device that uses a structured metallic Fresnel zone plate to strongly focus light. By combining a single-mode spliced to a piece of multimode fiber, an ultrathin platinum film, and FIB milling, a tailored Fresnel zone plate was realized on the fiber array and used for single cell trapping and detection.

X. Chen et al [link]: In this paper, the authors show the realization of an on-fiber ultra-flat diffractive lens by structuring a graphene layer. The key idea of that study relies on creating a tailored spatial phase distribution by laser-induced structuring of thin graphene layers located on a fiber end face, which the authors used to create a highly focused beam.

H. Ren et al [link]: The mentioned study is one of our previously published works on nanostructures on fibers and discusses achromatic light focusing using a nanoprinted fiber interfaced metasurface. It was already included in the submitted version of the manuscript.

B. Relevant: The following work is related to our manuscript, although no fiber interfacing was demonstrated.

J. He et al [link]: This paper presents a novel approach to the design of a superfocusing lens based on a Fresnel zone plate that employs higher orders of diffraction. The authors demonstrate that strong light focusing can be achieved without the need for subwavelength structures, making the concept attractive for potential adaptation to optical fibers. While the paper does not address fiber interfacing or applications in fiber photonics, the innovative nature of the approach has led us to include that work.

C. Not relevant: Although the following work shows a highly relevant application, it seems only marginally relevant to our study.

C. Yu et al [link]: The work mentioned by the Reviewer is a News & Views article in Light: Science & Applications reporting on an original paper that discusses an endoscopic approach using a metasurface-based structure combined with a coherent fiber bundle. Since there is no physical connection between the metasurface and the fiber bundle, and the implementation of such a structure on the fiber endface seems very challenging from our perspective, the connection to our work is minimal. Therefore, we have decided not to include this work in our manuscript and hope that the Reviewer is satisfied with the other additions made.

In summary, we have included the relevant works into the manuscript as follows: ... This approach is inline with the 'lab-on-fiber' concept [REFs] and has led to the development of complex fiber devices, including diffractive gratings [REFs], microlenses [REFs], metasurfaces [REFs] and meta-lenses [REFs]. **In addition to fabrication approaches such as modified electron beam lithography [link] or focused ion beam milling [link], this area of research has recently been boosted by the use of 3D nanoprinting technology [REFs], which, unlike conventional wafer-based lithographic techniques, is compatible with fiber geometry. ...**

... It is important to highlight that the fabrication process employed generally allows for the creation of structures with significantly greater complexity, thereby substantially broadening the spectrum of potential applications. **Here, advances in fiber end face functionalization (e.g., laser structuring of graphene layers [link], focused ion-beam milling [link], modified electron beam lithography [link]) or conceptually innovative approaches**

(e.g. focussing via higher-diffraction orders [link]) may open new opportunities to create complex on-fiber devices with unique properties, in particular for efficient light coupling. ...

R2.10: In Fig.5, a) is not defined in the description of the figure.

We thank the Reviewer for pointing out this missing label. We have changed the text of the caption of Fig. 5 as follows: ... Results of the optical characterization. **(a)** Measured coupling efficiencies of the individual cores containing nanostructures (colored curves) compared to a blank core (black line). ...

R2.11: In the paragraph below Table 2, some Figures are mentioned but it seems that the PDF file was not correctly compiled. This

We thank the Reviewer. The issue has been fixed in the new version of the manuscript: ... Applying the equations from steps 1 to 3 ..., with the corresponding angular distribution of the coupling efficiency shown by the green curves in Figs. **S4g-S4h**. ... Subsequent numerical optimization (step 4) produces the final optimized parameters ..., showing an exact overlap of the maximum coupling efficiency with the target angle of 40° (blue curves in Figs. **S4g-S4h**).

R2.12: In the first paragraph of section 4.2, the reference cited is not correctly compiled. This happens as well in section S3.

We thank the Reviewer for bringing this error to our attention. The issue has been fixed in the new version of the manuscript.

Response to comments of Reviewer #3

This study presents an angular demultiplexer achieved by applying nanoprinted periodic structures onto the facet of single-mode multicore fibers. Light with a large incident angle, significantly exceeding the acceptance angle of the single-mode fiber (SMF), can be redirected through diffraction at the periodic structure on the fiber's surface, enabling the light coupled into the core of single-mode multicore fibers. The author demonstrated the in-coupling efficiency exceeds 1% for incident angles between 20 to 50 degrees and 0.1% to 1% for angles between 50 and 80 degrees. This is an interesting result that deserves to be published. I have however several critical points that must be revised prior to publication.

We sincerely thank the Reviewer for the overall positive assessment of our work, the recommendation for publishing it, and for the thorough review of the manuscript. Below, we have provided detailed responses to each of the Reviewer's comments. We hope that our revisions have sufficiently addressed all concerns and contributed to a stronger and more refined version of the manuscript.

R3.1: The paper claims this device is "all-fiber integrated", which usually means the device splice robustly with other fiber components. i. e. Sohanpal, Ronit, et al. "All-fibre heterogeneously-integrated frequency comb generation using silicon core fibre." (2021). The device used in this work was fabricated by printing nano periodic structures on the facet of multicore fibers and has not been spliced with other fiber component. It is not sensible to use "all-fiber integrated".

We thank the Reviewer for raising the important point about appropriate wording. The Reviewer has a valid point that in fiber optics, "all-fiber integrated" typically implies that the device can be seamlessly spliced or connected to other fiber components, ensuring both mechanical and optical robustness. Since the device in this study involves nanostructure printing on the fiber facet without splicing to other fiber components, the term "all-fiber integrated" may indeed be misleading.

To address the Reviewer's comment, we have decided to replace the term "fiber-integrated" with "fiber-based" throughout the manuscript. This includes changing the title of the manuscript to: ... **Fiber-based angular demultiplexer using nanoprinted periodic structures on single-mode multicore fibers** ...

R3.2: It would be better to have schematic figures for the 3D nanoprinting and optical measurement procedure, which would produce more details and could be helpful for other labs to follow and reproduce this work after being published.

We would like to thank the Reviewer for the important advice on the visualization of the various experimental methods. We have taken this as an opportunity to produce appropriate schematic drawings and include them in the Supplementary Information as explained below.

A. Visualization of the 3D nanoprinting process

The direct laser writing process used to 3D nanoprint polymeric nanostructures on the endface of the multicore fibers is schematically shown in Fig. RL5 and consists of the

following steps:

Step 1 - Resin immersion: The printer, equipped with an inverted microscope (objective pointing upwards), is used for both laser printing and imaging. First, the resin (monomer, IP-dip 2) is applied to the objective and the fiber is fixed in a holder above it with the end facing down. The objective then moves upward, dipping the fiber into the resin and stopping when the fiber reaches the focal distance.

Step 2 - Laser writing: The polymer is cross-linked at desired locations using two-photon polymerization (2PP) with a moving femtosecond laser focus. Note that the 2PP process is essential to achieve nanoscale dimensions with direct laser writing due to the small polymerization volume of 2PP. The laser focus is moved relative to the fiber end face via scanning mirrors in the horizontal plane and by a piezo element in the vertical direction, allowing polymerization along its path and ultimately forming the nanostructures.

Step 3 - Development: Finally, the structure is developed by removing the unexposed resin with selected solvents (see the Methods section for details), yielding a sample that can be directly used in the optical experiments.

Fig. RL5 [Supplementary Figure 10]: Visualization of the individual steps of the 3D nanoprining process used in this work to create polymeric nanostructures on the end face of multicore fibers.

To address the Reviewer's comment, we have included Fig. RL5 and the related description in the Supplementary Information and referenced it in the main text as follows: ... The designed structures were realized on the MCFs discussed above according to the optimized parameters (Table 1) using 3D nanoprining (SEM images in Fig. 4). **The direct laser writing process used to 3D nanoprinate the structures on the MCF end face is described in Supplementary Note 6 (Supplementary Fig. 10).** ...

B. Visualization of the optical measurement setup

The optical experiments include a setup that consists of the following aspects (a schematic drawing of the setup, including a photographic image, is shown in Fig. RL6): At the input side, collimated light (IR-SLED, $\lambda_0 = 1550$ nm) was directed onto the nanostructure at various angles of incidence using a fiber-coupled collimator (Thorlabs F230FC-1550) to

achieve a collimation divergence of $<0.2^\circ$. The light was linearly polarized and controlled via a polarizer and a wave plate, yielding a total power of 10.5 mW incident on the sample. The fiber of interest was mounted on a rotary stage equipped with a high-precision XYZ stage (rotation accuracy: 0.5°), and the end face of the fiber with the nanostructures was precisely aligned with the axis of rotation. Note that the collimated beam was sufficiently large to uniformly illuminate the entire fiber end face. The output light from the opposite fiber end was imaged onto an infrared camera (320×256 pixels, pixel size: $30 \mu\text{m}$) using a $60\times/0.75$ objective.

Fig. RL6 [Supplementary Figure 11]: Visualization of the experimental setup used to measure the light coupling to different fiber cores. (a) Schematic diagram showing all relevant components (see main text for details). (b) Photographic top view of the setup with key components labeled.

To address the Reviewer's comment regarding the visualization of the measurement procedure, we have included this schematic of the setup (Fig. RL6) in the Supplementary Note 6 that discusses optical characterization. The text in this section has been modified accordingly: ... To measure the in-coupling efficiency, collimated light (IR-SLED, $\lambda_0 = 1550$ nm) was directed to nanostructure at various incident angles and the light output from the opposite end of the fiber was imaged on a camera (a schematic drawing of the setup, including a photographic image is shown in Supplementary Fig. 11). ...

The main text has been modified as follows: ... Optical measurements were performed by illuminating the nanostructured end of the sample with collimated light ($\lambda_0 = 1550$ nm) and capturing the output pattern with a camera (c.f., Supplementary Note 6 and Supplementary Fig. 11). ...

C. Visualization of the optical measurement procedure

Data acquisition is performed as follows: (1) adjustment of the angle of incidence, (2) fine adjustments of the light beam to maximize the total power at the output, (3) selection of appropriate neutral density (ND) filters and exposure time to obtain the maximum unsaturated signal on the camera, and (4) acquire the images. To increase the dynamic range, multiple images were taken for each angle of incidence as well as for each core using different combinations of exposure time and ND filters, including images recorded next to the

cores as a reference to take the camera detector noise into account, allowing to precisely determine the power in each core for all incidence angles. This procedure can be visualized in the following sequence (Fig. RL7).

Fig. RL7 [Supplementary Figure 12]: Visualization of the measurement procedure for the acquisition of the power distribution over the cores at the output of the nanostructure-empowered MCF.

To address the Reviewer's comment regarding the visualization of the measurement procedure, we have added this text and the new schematic (Fig. RL7) to the Supplementary Note 6: ... Since the intensity changes significantly between individual cores and over the angular range of interest (in some cases by orders of magnitude), a separate image was recorded for each core with an angular increment of 1° . The exposure time of each image was adjusted to maximize the signal on the camera sensor without causing oversaturation. In addition, different neutral-density filters were used to adjust the intensity. **Details of the measurement procedure including a visualization can be found in Supplementary Fig. 12.** ...

Also, the main text has been slightly modified: ... Optical measurements were performed by illuminating the nanostructured end of the sample with collimated light ($\lambda_0 = 1550$ nm) and capturing the output pattern with a camera (c.f., Supplementary Note 6). To compensate for the limited dynamic range of the camera, images were taken with several exposure times at the same angle if necessary (**the related measurement procedure is shown in Supplementary Fig. 12**). The angular distribution of the coupling efficiency was determined by rotating the MCFs, mounted on a high-precision rotary stage (accuracy is 0.5°), relative to the input beam. ...

R3.3: In Figure 6: why in the beam in the fig of 25 degree is brighter than 30 degree and the fig of 35 degree is brighter than 40 degree. Is this caused by the normalization to a maximum signal-to-noise ratio at the specific angle of incidence? It would be better to use the same normalization level, and consistent with the video this paper produced.

We thank the Reviewer for pointing out the different brightness of the images. The brightness variation is due to different camera settings during image acquisition. Due to the relatively limited dynamic range of the used camera (8-bit), multiple images were taken for each angle of incidence to compensate for either too low intensity or image saturation. This procedure has already been mentioned in the Methods Section in the text and is the reason for the different brightness of the images in Fig. 6.: ... Optical measurements were performed by illuminating the nanostructured end of the sample with collimated light ($\lambda_0 = 1550$ nm) and capturing the output pattern with a camera (c.f., Supplementary Information S4). To compensate for the limited dynamic range of the camera, images were taken with

several exposure times at the same angle if necessary. The angular distribution of the coupling efficiency was determined by rotating the MCFs, mounted on a high-precision rotary stage (accuracy is 0.5°), relative to the input beam. ...

To address the Reviewer's comment, we have calculated the different gain factor for all images relative to the image at normal incidence ($\theta = 0^\circ$) and summarized them in the following table, which is included in the Supplementary Note 6:

...

Amplification factor. The variation in brightness of the images shown in Fig. 6 is due to the different camera settings used during image acquisition. Due to the relatively limited dynamic range of the camera used (8-bit), multiple images were taken for each angle of incidence to compensate for either low intensity or image saturation. The corresponding gain factors for all images relative to the image at normal incidence ($\theta = 0^\circ$) are given in Supplementary Table 4.

Tab. RL 4 [Supplementary Table 4]: Gain factors used for the images shown in Fig. 6 to compensate for either low intensity or image saturation. Note that the middle column indicates the core for which the image was optimized. In this context, these cores are specified by the design angle of incidence. For example, a value of 50° of the middle column refers to the core in the lower right corner shown in the images of Fig. 6.

incident angle	optimized core (number refers to the respective design angle)	gain factor
0°	30°	1
10°	30°	113
20°	20°	3
25°	20°	227
30°	30°	6.2
35°	20°	113
40°	40°	8.1
45°	50°	38
50°	50°	11.3
55°	60°	47
60°	60°	20
65°	70°	47
70°	70°	47
75°	80°	113
80°	80°	151

...

The text of the caption of Fig. 6 has been changed accordingly: ... Selected measured images of the output intensity of the nanostructured-enhanced multicore fiber (Video 1), showing the power distribution across the different cores at different angles of incidence (given at the bottom of each image). In each image, the domains of the fiber cores are indicated by dashed cyan circles. The optimised angles θ_{-1} , in unit of degree) together with the associated core are indicated in the schematic on the left. **Note that each image was captured with a different gain factor to compensate for the limited dynamic range of the camera used (a table of the corresponding gain factors is provided in**

Supplementary Note 6). ~~The brightness of each picture is normalized to a maximum signal to noise ratio at the specific angle of incidence. ...~~

R3.4: Section 4.1 “the green curves in Figs. ??g-??h ” and “blue curves in Figs. ??g-??h ” please fix the editing errors.

We thank the Reviewer for bringing this error to our attention. The issue has been fixed in the new version of the manuscript: ... Applying the equations from steps 1 to 3 ..., with the corresponding angular distribution of the coupling efficiency shown by the green curves in Figs. **S4g-S4h**. ... Subsequent numerical optimization (step 4) produces the final optimized parameters ..., showing an exact overlap of the maximum coupling efficiency with the target angle of 40° (blue curves in Figs. **S4g-S4h**).

Response to Reviewer #2

The Authors have successfully addressed thoroughly and meticulously all the points after the first revision. The additional material and results included in the new manuscript resulted in a nice work that deserves to be published. I thank the Authors for extending the experimental work and simulations to complement the submitted manuscript. As a result, the new version of the manuscript reads very well and explains in detail the work. Also, as previously mentioned, the quality and importance of the work is very high. Therefore, I fully recommend that the article be published. I only have a few comments.

We sincerely thank the Reviewer for the thorough review of our manuscript and the detailed comments that helped us to improve the manuscript. We are grateful for him/her high evaluation of our work.

The addition of Supplementary Figure 4 is a nice addition to the content of the manuscript. A suggestion that will improve the Figure is to also illustrate with dashed-line circles where the other cores should be; this will also help to avoid misunderstandings due to the artifact visible in the images.

We thank the Reviewer for this suggestion. Following this advice, we added the white-line circles corresponding to the core positions and the outer diameter of the multicore fiber in Supplementary Figure 4.

In the revised manuscript, I observed on page 14 that the Authors mention Figs. S4g-S4h which I believe are incorrect. After the new additions to the manuscript it is now S5g-S5h. As new material (tables and figures) were added, please check the numbering and mentions of those.

We thank the Reviewer for this observation. This part was fixed and transferred to the Supplementary Information following the editorial requests.

The first part of the question previously done (R2.2) the intention was more to know if the distribution of the different structures had a specific order. Regarding the design of the structures, the original version of the manuscript was already well described. Nonetheless, with the information added regarding the questions from reviewer 1, it is clear that the light is intended to impinge on the whole fiber facet of the fiber. Thus, this is clarified.

We thank the Reviewer for this comment. As he/she mentioned, we add this sentence into the "Optical measurement procedure" of the Methods section as: ... Note that the entire fiber end face is illuminated. ...

Response to all Reviewers

We thank the Reviewers again for their careful and meticulous work that allowed us improving and clarifying important points of our manuscript.